# Constructing robust heterostructured interface for anode-free zinc batteries with ultrahigh capacities

Xinhua Zheng[1,3], Zaichun Liu ●[1,3], Jifei Sun[1], Ruihao Luo[1], Kui Xu[1], Mingyu Si[2], Ju Kang[2], Yuan Yuan[1], Shuang Liu[1], Touqeer Ahmad[1], Taoli Jiang[1], Na Chen[1], Mingming Wang[1], Yan Xu[1], Mingyan Chuai[1], Zhengxin Zhu[1], Qia Peng[1], Yahan Meng[1], Kai Zhang[1], Weiping Wang[1] & Wei Chen ●[1] ✉

The development of Zn-free anodes to inhibit Zn dendrite formation and modulate high-capacity Zn batteries is highly applauded yet very challenging. Here, we design a robust two-dimensional antimony/antimony-zinc alloy heterostructured interface to regulate Zn plating. Benefiting from the stronger adsorption and homogeneous electric field distribution of the $Sb/Sb_2Zn_3$-heterostructured interface in Zn plating, the Zn anode enables an ultrahigh areal capacity of 200 mAh cm$^{-2}$ with an overpotential of 112 mV and a Coulombic efficiency of 98.5%. An anode-free Zn-Br$_2$ battery using the $Sb/Sb_2Zn_3$-heterostructured interface@Cu anode shows an attractive energy density of 274 Wh kg$^{-1}$ with a practical pouch cell energy density of 62 Wh kg$^{-1}$. The scaled-up Zn-Br$_2$ battery in a capacity of 500 mAh exhibits over 400 stable cycles. Further, the Zn-Br$_2$ battery module in an energy of 9 Wh (6 V, 1.5 Ah) is integrated with a photovoltaic panel to demonstrate the practical renewable energy storage capabilities. Our superior anode-free Zn batteries enabled by the heterostructured interface enlighten an arena towards large-scale energy storage applications.

The coupling of high-safety, low-cost, and sustainable battery systems with electrical grid is proved effective to assure continual energy supply and overcome the intermittent nature of renewable energy[1–8]. Aqueous Zn batteries have received much attention in terms of high theoretical energy density, high safety, and their sustainable raw materials with low-cost[9–14]. However, Zn anode suffers from unfavorable reactions, particularly the formation of Zn dendrites, which lead to low Coulombic efficiency (CE) and even short-circuiting of the battery in prolonged cycles[15,16]. Moreover, the excessive use of Zn in typical Zn batteries causes the negative to positive (N/P) ratio goes as high as dozens or even hundreds, which earnestly declines the practical energy densities of the batteries and poses a severe threat to their real-life application[17–22]. It is important

to develop advanced strategies on inhibiting dendrite formation and controlling the over-utilization of Zn to realize high energy density and long service life Zn batteries[11,23–26].

Using well-engineered zincophilic materials such as Sn, Pb, Au, Ag, Cu, and C on anode current collectors can effectively inhibit the formation of Zn dendrites, while making them feasible to achieve anode-free Zn battery design[27–33]. This design essentially controls the N/P ratio and achieves zero excess of anode in the battery, which not only enhances energy density of the battery via lowering its volume and weight, but also simplifies the battery construction process to reduce cost[22,34]. To accomplish the anode-free operation, anodic active materials in the form of dissolvable ions in the electrolyte are transformed into metal on the anode current collector upon charge and dissolved back

[1]Department of Applied Chemistry, School of Chemistry and Materials Science, Hefei National Research Center for Physical Sciences at the Microscale, University of Science and Technology of China, 230026 Hefei, Anhui, China. [2]School of Mechanical Engineering, Beijing Institute of Petrochemical Technology, 102617 Beijing, China. [3]These authors contributed equally: Xinhua Zheng, Zaichun Liu. ✉e-mail: weichen1@ustc.edu.cn

into electrolyte during discharge. Such a cutting-edge strategy has been exploited in Zn batteries, which showed encouraging characteristics[33,35–38]. Recently, Cui and coworkers[33] designed a C@Cu current collector for Zn plating/stripping, achieving a high average CE of 99.6% over 300 cycles at an areal capacity of 0.5 mAh cm$^{-2}$. Feng and coworkers[16] proposed a 3D Ti-TiO$_2$ substrate for dendrite-free Zn battery, in which the Zn|3D Ti-TiO$_2$ cell cycled stably for 200 h at 5 mAh cm$^{-2}$ with a CE of ~93.7%. However, deep understanding of the mechanisms for anode-free Zn electrodeposition and the development of Zn anode towards high capacities for practical Zn battery applications have not been achieved.

The typical process of Zn electrodeposition on an anode-free substrate can be divided into three consecutive stages, i.e., Zn nucleation, nucleus growth and further Zn growth[23]. Amongst, the homogeneous Zn nucleation is one of the most critical and decisive steps for achieving the dendrite-free Zn electrodeposition. It can be ascribed to the mechanism that numerous uniform nucleation sites help guide the subsequent growth of the nucleus to maintain its uniformity and ultimately form a uniform plating layer. Therefore, designing a robust, zincophilic and low-cost interface to regulate the Zn nucleation on anode-free Zn substrates is highly desirable[23,31,39]. For instance, Zhi and coworkers[30] demonstrated the quasi-isolated Au particles as heterogeneous seed to guide uniform Zn electrodeposition, and the optimized Au@Zn anode presented a cycling stability of over 2000 h at an areal capacity of 0.05 mAh cm$^{-2}$. Dong and coworkers[32] designed a zincophilic Cu nanowire to stabilize Zn anode, where the Cu substrate effectively inhibited Zn dendrite growth for up to 130 h at an areal capacity of 5 mAh cm$^{-2}$. Despite the effectiveness of the interface engineering for stable Zn electrodeposition, the attained areal capacity and lifespan of the Zn batteries are inadequate for industrialization (Supplementary Table 1). Further efforts should be made to develop robust interfaces on anode-free substrates for stable Zn electrodeposition and stripping at high capacities, which is one of the key challenges towards the high-energy Zn batteries.

Herein, we propose a two-dimensional (2D) heterostructured interface to regulate the homogeneous Zn nucleation and growth, which results in dendrite-free Zn electrodeposition and stripping with ultrahigh capacities. An antimony/antimony-zinc alloy heterostructured interface (Sb/Sb$_2$Zn$_3$-HI) is used as a model. Due to its strong adsorption for Zn atoms and homogeneous electric field in Zn plating, the Sb/Sb$_2$Zn$_3$-HI helps reduce Zn nucleation barriers and consequently regulate homogeneous Zn nucleation and growth. The dendrite-free Zn deposition in an ultrahigh areal capacity of 200 mAh cm$^{-2}$ can be achieved on the Sb/Sb$_2$Zn$_3$-HI coated Cu foil (Sb/Sb$_2$Zn$_3$-HI@Cu) with low overpotentials and high CEs. The fabricated anode-free Zn–Br$_2$ battery using the Sb/Sb$_2$Zn$_3$-HI@Cu anode shows much-improved energy density and cycling stability than its counterparts without the HI layer. Particularly, the Zn–Br$_2$ battery is scaled up to an Ah capacity level and displays high energy density for the practical battery storage module by charging from photovoltaic cells. Our anode-free design strategy will enable a revolution towards the commercialization of Zn batteries for large-scale energy storage applications.

## Results

### Design of 2D heterostructured interface

The 2D heterostructured interfaces are constructed by low-cost, zincophilic metal and its alloy with Zn, which provides a fertile ground for nucleating Zn[29–31,40–42]. We selected antimony (Sb) metal in this study as a model to demonstrate the design strategy of the 2D heterostructured interfaces owing to the relatively low-cost (~10 US\$ kg$^{-1}$) of Sb and its rich alloying phases with Zn[43,44]. Cu was selected as the substrate for the Zn-free anode to build up a durable Sb@Cu electrode. As illustrated in Supplementary Fig. 1, a typical Cu$_2$Sb phase was detected in the XRD results, which indicates that Sb has a certain solubility in Cu,

promoting a strong bond between the two metals and thus establishing a robust Sb-Cu interface[45]. We firstly evaluated the stability of the prepared Sb@Cu substrates in aqueous electrolytes. As shown in Supplementary Fig. 2, the Tafel plots indicate good stability of the Sb@Cu substrate with a corrosion current density of 0.049 mA cm$^{-2}$, which is well below 0.21 mA cm$^{-2}$ for Zn foil and 0.072 mA cm$^{-2}$ for Cu foil. When the Sb@Cu electrode was plated by Zn, a antimony/antimony-zinc alloy heterostructured interface (Sb/Sb$_2$Zn$_3$-HI) was formed and its corrosion current density was further reduced to 0.025 mA cm$^{-2}$, which suggests good stability of the 2D heterogeneous interface formed by Sb and Sb$_2$Zn$_3$ alloy. Furthermore, the Sb/Sb$_2$Zn$_3$-HI@Cu electrode exhibits the lowest hydrogen evolution reaction overpotential of −1.60 V vs. RHE at 10 mA cm$^{-2}$, which is lower than that of Zn foil (−1.38 V vs. RHE), Cu foil (−1.27 V vs. RHE) and Sb@Cu (−1.46 V vs. RHE) (Supplementary Fig. 3), implying the improved stability of the Sb/Sb$_2$Zn$_3$-HI@Cu electrode towards HER inhibition.

### Working mechanism of Zn electrodeposition on Sb/Sb$_2$Zn$_3$-HI@Cu electrode

Figure 1a exhibits schematic diagrams of the Zn electrodeposition on Zn foils and Sb@Cu substrates. The typical Zn electrodeposition on Zn foil results in the initial inhomogeneous nucleation, then the growth of nuclei and intensification of inhomogeneity, and eventually uncontrolled dendrite formation. In contrast, on the Sb@Cu electrode, the initially plated Zn can spontaneously alloy with Sb to form the Sb/Sb$_2$Zn$_3$-HI[31,44], thus leading to homogeneous Zn nucleation in the initial Zn electrodeposition. Subsequently, the homogeneous nucleus gradually grows and finally a dendrite-free and homogeneous galvanized layer is formed. Different tests were carried out to identify this mechanism intuitively. Figure 1b displays the voltage profiles of Zn electrodeposition upon various substrates including Zn, Cu and Sb@Cu. On the Sb@Cu substrate, the lower nucleation overpotential of 3.6 mV is recorded, which is well below that of 5.1 mV for Zn foil and 9.3 mV for Cu foil. This can be ascribed to the promoted Zn nucleation on the zincophilic Sb/Sb$_2$Zn$_3$-HI.

To probe the chemical composition of the Sb@Cu substrate after Zn electrodeposition, X-ray diffraction (XRD), X-ray photoelectron spectroscopy (XPS) and transmission electron microscope (TEM) have been utilized. As illustrated in Fig. 1c, the XRD pattern assured the existence of Sb$_2$Zn$_3$, which corresponds to JCPDS # 00-023-1016, reflecting the formation of Sb$_2$Zn$_3$ alloy at the initial Zn electrodeposition stage. In addition, the crystalline phase of Sb (JCPDS # 01-071-1173) is also detected in the same sample, indicating the successful generation of Sb/Sb$_2$Zn$_3$-HI. When the electrodeposited Zn over the Sb@Cu substrate reaches a high capacity of 10 mAh cm$^{-2}$, the characteristic peaks of metallic Zn are significantly enhanced, which accompanied by the disappearance of the alloy phase. Interestingly, after the complete dissolution of Zn, the Sb$_2$Zn$_3$ alloy phases are still detected on the Sb@Cu substrate. This implies that the Sb/Sb$_2$Zn$_3$-HI formed in the initial Zn electrodeposition can retain on the Sb@Cu surface and continue to regulate the Zn nucleation in the subsequent cycles. XPS results of the Sb@Cu substrate exhibit that the peak of Zn 2p shifted about 0.2 eV towards the lower binding energy when the electrodeposition capacity of Zn increased from 0.2 mAh cm$^{-2}$ to 10 mAh cm$^{-2}$ (Fig. 1d)[46,47]. This can be attributed to the effect of the more electronegative Sb on Zn in the Sb$_2$Zn$_3$ alloy than the pristine Zn deposit. Furthermore, high-resolution TEM confirms the coexistence of (213) crystal planes for Sb$_2$Zn$_3$ and (104) for Sb of the Sb@Cu substrate after the initial Zn plating (Fig. 1e and Supplementary Fig. 4a–d), while the EDS-mapping also displays the stacked distribution of Sb and Zn in their interfacial areas that is ascribed to the Sb$_2$Zn$_3$ alloy (Fig. 1f). All the above characterization results suggest the successful generation of the Sb/Sb$_2$Zn$_3$-HI@Cu electrode.

The DFT calculations and COMSOL simulations were used to further understand the effects of the Sb/Sb$_2$Zn$_3$-HI on Zn

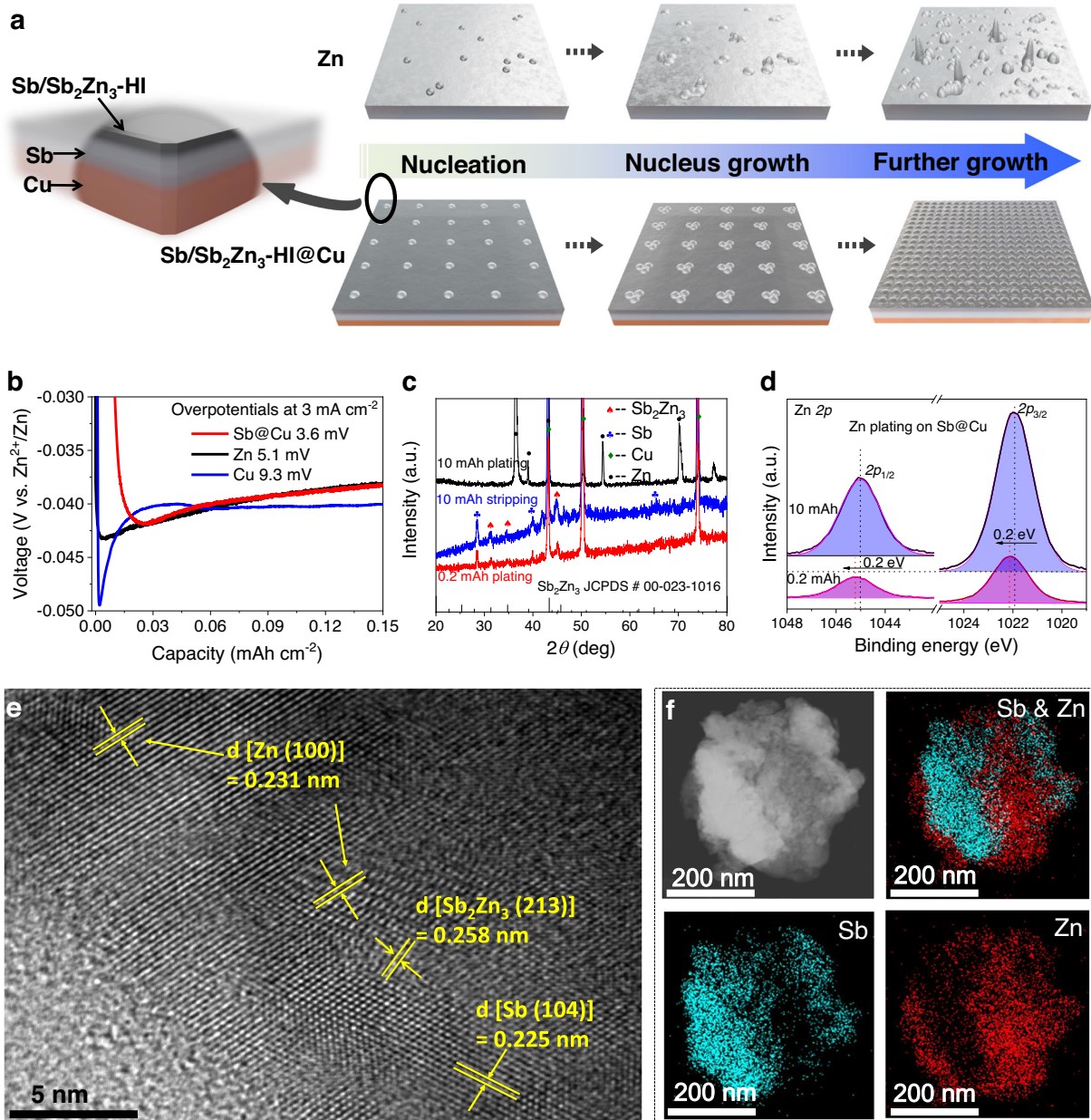

**Fig. 1 | Mechanism of the anode-free Zn electrodeposition. a** Schematic diagrams of Zn electrodeposition on Zn and Sb/Sb$_2$Zn$_3$-HI@Cu substrates (HI represents heterostructured interface). **b** Nucleation barriers of the Zn electrodeposition on Zn, Cu and Sb@Cu substrates, where the charge current density is 3 mA cm$^{-2}$ in 2 M ZnBr$_2$. **c** XRD patterns of the Zn electrodeposition and stripping on Sb@Cu at different capacities. **d** XPS spectra of Zn 2p of Zn plating on Sb@Cu substrate with different capacities. **e** HRTEM images of the Sb$_2$Zn$_3$-HI@Cu anode. **f** EDS-mapping of the Zn and Sb in the Sb$_2$Zn$_3$-HI@Cu anode. The sample for HRTEM and EDS-mapping measurements was prepared by plating 0.2 mAh Zn on Sb@Cu substrate. The constant current electrodeposition shown in Fig. 1b was carried out at room temperature (25 °C).

electrodeposition. To assess the affinity of Sb with Zn, we studied the adsorption behaviors of Zn atoms on both Zn and Sb@Cu substrates, where the crystal planes of Zn (100) and Sb (104) were chosen according to the XRD and TEM results (Fig. 1e and Supplementary Fig. 5). As shown in Fig. 2a, the adsorption energy of Zn atoms on Sb (104) is −3.12 eV, which is much lower than that of Zn (100) plane (−0.72 eV). The stronger adsorption ability of Sb for Zn atoms can be ascribed to the initial Zn/Sb alloying. As illustrated in Supplementary Figs. 6, 7 and Table 2. The Sb@Cu substrate displays an ohmic resistance of 1.91 Ω, which is lower than 2.56 Ω of the Zn foil. Meanwhile, it also exhibits a lower charge transfer resistance of 2.62 Ω than 19.78 Ω of the Zn foil, indicating that the low adsorption energy on the Sb surface can effectively accelerate the charge transfer. Ensuring the

continuous strong adsorption of Zn atoms on the substrate surface is the key to achieve high areal capacity. Thus, we further evaluated the adsorption energy of Zn atoms on the Sb surface after the initial Zn adsorption. It can be assumed that the subsequent adsorption of Zn atoms took place on the Sb/Sb$_2$Zn$_3$-HI, since the Sb$_2$Zn$_3$ alloy layer was formed after the initial Zn electrodeposition. The adsorption of Zn atoms by the Sb/Sb$_2$Zn$_3$-HI maintains a lower adsorption energy value than the Zn for many layers of Zn deposits. In addition, the geometric information clearly present that a stable Sb/Sb$_2$Zn$_3$ interface is gradually formed with the increase of the adsorbed Zn number (Supplementary Fig. 4e). We further calculated the structure of Sb adsorbed with 21 Zn atoms and its structural configuration indicated that the formed Sb/Sb$_2$Zn$_3$ interface layer was more stable as the number of

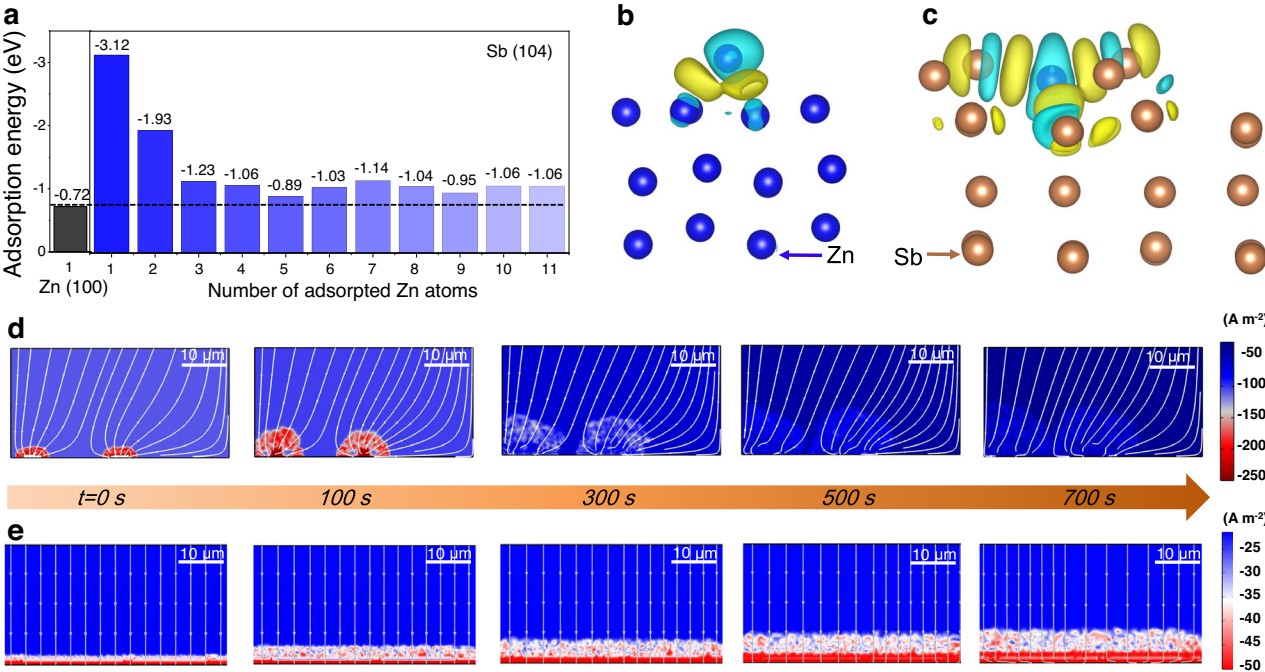

**Fig. 2 | DFT calculations and COMSOL simulations. a** Adsorption energy of the Zn atoms on Zn (100) and Sb (104) crystal planes. Theoretical calculation models with differential charge density of **b** Zn (100) and **c** Sb (104). **d** Simulated current density distribution of Zn plating on Zn substrate. **e** Simulated current density distribution of Zn plating on Sb/Sb$_2$Zn$_3$-HI@Cu substrate. The simulation processes were carried out at room temperature (25 °C).

adsorbed atoms increases. This indicates that Zn deposition on Sb/Sb$_2$Zn$_3$-HI is energetically favored, verifying the positive effect of the Sb/Sb$_2$Zn$_3$ interfacial layer on the homogeneous Zn electrodeposition[48–50]. As revealed by differential charge density plots in Fig. 2b, c, the electrons around the deposited Zn atoms on the Sb surface are more delocalized than that on the Zn surface. This helps to enhance the metal bonding between Zn atoms and the Sb@Cu electrode and thus facilitates the homogeneous deposition of Zn.

Moreover, "COMSOL Multiphysical" was used to simulate the Zn plating process on Zn and Sb/Sb$_2$Zn$_3$-HI@Cu substrates. When plating on Zn surface, the dynamic current density distribution diagrams (0–700 s) show a persistent irregularity throughout the deposition process, resulting in the gradual formation of dendrites (Fig. 2d). This can be attributed to the high nucleation barriers on the Zn surface, leading to inhomogeneous nucleation and charge aggregation near the Zn nuclei (red regions in Fig. 2d), eventually to non-uniform Zn electrodeposition. Further evidence can be obtained from the Zn$^{2+}$ ions flux distribution on the Zn surface (Supplementary Fig. 8a), where Zn$^{2+}$ ions tend to occur in areas of charge aggregation, leading to inhomogeneous Zn electrodeposition. In contrast, the Sb/Sb$_2$Zn$_3$-HI exhibits a homogeneous electric field and an even Zn$^{2+}$ flux distribution throughout the dynamic Zn electrodeposition process, which leads to homogeneous Zn nucleation on the Sb/Sb$_2$Zn$_3$-HI and consistently promotes uniform Zn deposition (Fig. 2e and Supplementary Fig. 8b). Overall, the simulation results further reinforce the dendrite-free Zn electrodeposition mechanism, in which the homogeneous Zn electrodeposition is spurred by a uniform Zn$^{2+}$ flux and electric field distribution.

## Morphology of Zn electrodeposition on Zn and Sb/Sb$_2$Zn$_3$-HI@Cu substrates

To further visually observe Zn electrodeposition on the Zn and Sb/Sb$_2$Zn$_3$-HI@Cu substrates, scanning electron microscope (SEM) was used to assess the morphology of electrodeposited Zn with different areal capacities varied from 1 to 50 mAh cm$^{-2}$. As shown in Fig. 3a, the

inhomogeneous galvanic morphology on the Zn surface appears at the onset of electrodeposition (1 mAh cm$^{-2}$), which can be traced back to a non-uniform distribution of Zn nucleation sites leading to stronger tendency for Zn electrodeposition in local areas. Successive Zn electrodeposition accumulated in such inhomogeneous areas results in the pronounced protrusion as the capacity increases to 10 or even 50 mAh cm$^{-2}$ (Fig. 3b, c). We have also deposited Zn on Cu substrates at areal capacities varied from 1 to 100 mAh cm$^{-2}$ (Supplementary Fig. 9), where the surface morphology of Cu has distinct bulges that became more pronounced with increased capacity, which share the similar behavior as the Zn substrate. However, a homogeneous and robust Zn plated layer is obviously observed on the Sb@Cu substrate at an areal capacity of 1 mAh cm$^{-2}$, which can be ascribed to the homogeneous Zn nucleation on Sb/Sb$_2$Zn$_3$-HI that results in a uniform Zn electrodeposition (Fig. 3d). This homogeneous galvanized layer can be extended to high areal capacities of 10 and 50 mAh cm$^{-2}$ (Fig. 3e, f), indicating the regulation effect continues throughout the entire Zn electrodeposition process. In-situ observation of Zn electrodeposition morphology on Zn foil and Sb@Cu substrates in 2 M ZnBr$_2$ further demonstrates the effect of Sb/Sb$_2$Zn$_3$-HI on stable Zn electrodeposition (Supplementary Fig. 10). The Zn plating on Zn foil shows a non-uniform morphology in the initial stage (5 s), and this inhomogeneity persists as the Zn deposition time increases (25–500 s). After deposition up to 1000 s, some obvious bumps were formed and gradually increased with time. In contrast, the Zn electrodeposition on the Sb/Sb$_2$Zn$_3$-HI@Cu surface shows a homogeneous morphology, which is maintained throughout the entire galvanizing process (5–2000 s). It can be attributed to the homogeneous electric field kept at the Sb/Sb$_2$Zn$_3$-HI and thus driving the formation of a uniform galvanized Zn deposition layer. In addition, the cross-sectional morphology of Zn electrodeposition on the Sb@Cu substrate with an areal capacity of 10 mAh cm$^{-2}$ exhibits a dense and coherent morphology (Fig. 3g). The uniform distributions of Zn, Sb, and Cu are also confirmed by EDS-mapping. It is worth mentioning that the Sb@Cu substrate was optimized in a two-electrode configuration at 3 mA cm$^{-2}$ for 10 min,

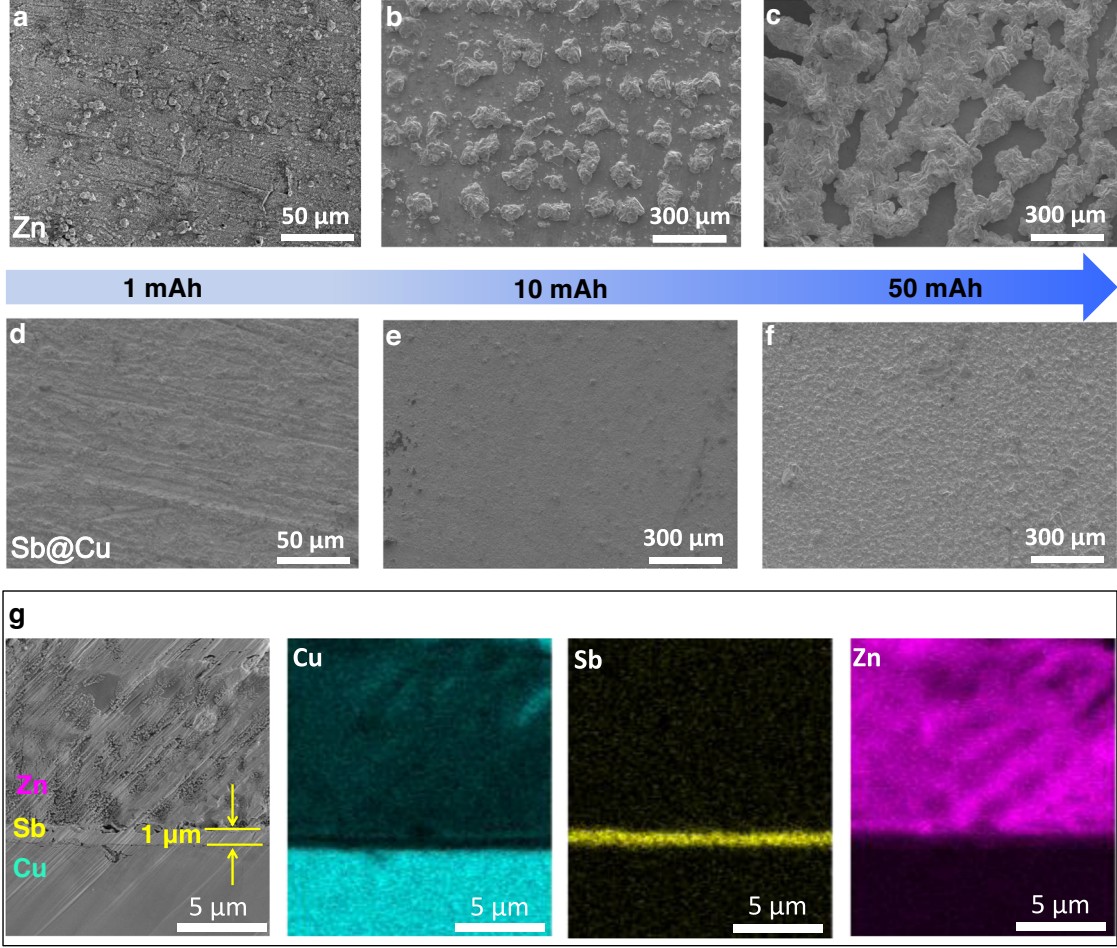

**Fig. 3 | Microstructures of the Zn electrodeposition on Zn and Sb@Cu substrates with different capacities.** Zn electrodeposited on Zn substrate with areal capacities of **a** 1 mAh cm$^{-2}$, **b** 10 mAh cm$^{-2}$, and **c** 50 mAh cm$^{-2}$. Zn electrodeposited on Sb@Cu substrate with areal capacities of **d** 1 mAh cm$^{-2}$, **e** 10 mAh cm$^{-2}$, and **f** 50 mAh cm$^{-2}$. **g** Cross-sectional SEM image and the corresponding EDS-mapping of Zn electrodeposition on Sb@Cu with a capacity of 10 mAh cm$^{-2}$.

resulting in Sb nano-spheres with a thickness of roughly 1 μm (Fig. 3g and Supplementary Fig. 11). Overall, the Sb@Cu substrate prepared by such an approach is proved successful in realizing uniform Zn plating, unfurling a strategy for dendrite-free Zn electrodeposition with high capacities.

### Electrochemical performance of Zn half-cells

The plating/stripping tests of various substrates including Zn, Cu and Sb@Cu were carried out in coin cells at 5 mAh cm$^{-2}$ and 5 mA cm$^{-2}$, with a Zn foil as counter electrode (Supplementary Fig. 12). Benefiting from the highly stable Zn electrodeposition on Sb/Sb$_2$Zn$_3$-HI, the Zn|Sb@Cu half-cell exhibits a superior stability over 700 h with a low overpotential of ~20 mV and an average CE of ~97.8%. In contrast, the polarization of the Zn|Zn symmetric cell begins to rise after 200 h of cycling and short-circuits after 360 h, while the Zn|Cu asymmetric cell fails after 240 h of cycling. The failure mechanisms can be attributed to the gradually accumulated Zn dendrites on the electrode and their eventual penetration through the separator, resulting in a cell breakdown. We further enlarged the areal capacity to 10 mAh cm$^{-2}$, wherein the Zn|Sb@Cu half-cell in a beaker configuration can be cycled at 20 mA cm$^{-2}$ for nearly 550 h with a high average CE of ~98% (Fig. 4a). On the contrary, the Zn|Zn half-cell can only operate for roughly 100 h, owing to the exacerbated non-uniform Zn electrodeposition at high areal capacities. Further, as shown in the voltage vs. capacity curves at the 10th, 100th and 300th cycles, the Zn|Sb@Cu cell maintains lower overpotentials than that of the Zn|Zn cell, along with high CEs (Fig. 4b).

Figure 4c portrays the rate capacities of the Zn|Sb@Cu half-cell, which displays a consistently high CE over 98% at 10 mAh cm$^{-2}$, even at a current density up to 200 mA cm$^{-2}$ (20 C). This implies that the advantages of Sb/Sb$_2$Zn$_3$-HI to stabilize Zn electrodeposition are not confined to high current densities, signaling its great potential for high-power Zn batteries. Furthermore, after 50 cycles at 10 mAh cm$^{-2}$, the morphologies of the Zn and Sb@Cu substrates disclose evidence of the advantages of Sb/Sb$_2$Zn$_3$-HI for Zn plating/stripping (Fig. 4d). With a substantial percentage of dendritic Zn, the Zn foil shows an irregular structure. In contrast, the Sb@Cu substrate shows a homogeneous and compact surface without any discernible dendrites, signifying that the Sb/Sb$_2$Zn$_3$-HI enables the efficient Zn plating/stripping throughout the prolonged cycling. Even when the areal capacity of the Zn|Sb@Cu half-cell is increased to 50 mAh cm$^{-2}$, the battery can still cycle stably for more than 220 h and maintain an average CE of 98.3% at 50 mA cm$^{-2}$ (Fig. 4e). The highly reversible anode-free Zn plating/stripping regulated by the Sb/Sb$_2$Zn$_3$-HI at high capacities makes Zn batteries commercially viable.

### Electrochemical performance of anode-free Zn–Br$_2$ battery

In the light of the attractive Zn plating/stripping performance of Sb/Sb$_2$Zn$_3$-HI, we carried out Zn-free full cell investigations of the Sb@Cu anode combined with the Br$_2$ cathode (Supplementary Fig. 13). The Zn–Br$_2$ batteries were assembled using Sb@Cu as the anode, carbon felt (CF) as the cathode, and low cost ZnBr$_2$ and tetrapropylammonium bromide (TPABr) as the electrolyte (Fig. 5a). As illustrated in Eqs. (1–3),

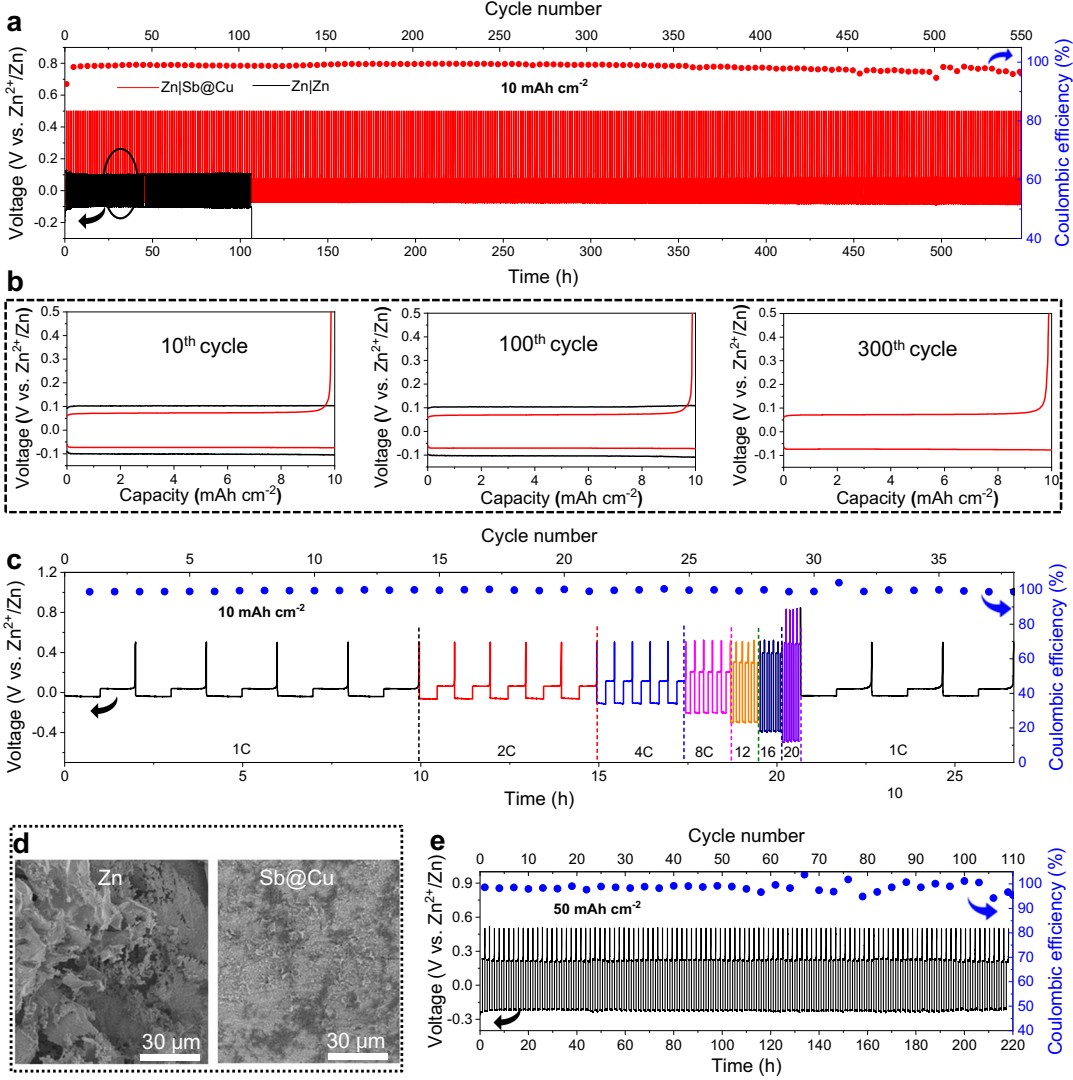

**Fig. 4 | Electrochemical performance of the Zn|Zn and Zn|Sb@Cu half-cells.**
**a** Cycling performance of the half-cells tested under an areal capacity of 10 mAh cm$^{-2}$ and a current density of 20 mA cm$^{-2}$, where the asymmetric Zn|Sb@Cu cell was set to a cut-off voltage of 0.5 V vs. Zn$^{2+}$/Zn. **b** Voltage vs. capacity curves at different cycles. **c** Rate capacities of the Zn|Sb@Cu half-cell at an areal capacity of 10 mAh cm$^{-2}$ and current densities from 10 to 200 mA cm$^{-2}$, where the cut-off voltages were set to 0.5 V vs. Zn$^{2+}$/Zn at 1–16 C and 0.8 V vs. Zn$^{2+}$/Zn at 20 C. **d** SEM images of Zn and Sb@Cu substrates after 50 cycles in the half-cells with an areal capacity of 10 mAh cm$^{-2}$ and a current density of 20 mA cm$^{-2}$. **e** Cycling performance of the Zn|Sb@Cu half-cell with an areal capacity of 50 mAh cm$^{-2}$, a current density of 50 mA cm$^{-2}$, and a cut-off voltage of 0.5 V vs. Zn$^{2+}$/Zn. The electrochemical measurements were carried out at room temperature (25 °C).

when charging the battery, soluble Br$^-$ in the electrolyte diffuses to the cathode and is oxidized to Br$_2$, followed by complexes with TPABr to form TPABr$_3$ on the CF cathode, while Zn$^{2+}$ is reduced to plate Zn at the Sb@Cu anode. During discharge, the as-deposited TPABr$_3$ on the cathode is dissolved back to soluble Br$^-$ and TPA$^+$, and Zn is stripped from the anode[9]. In this advanced battery design, the active materials are completely dissolved into the electrolyte and thus the electrodes do not require any pre-treatment, which greatly simplifies the cell preparation process and reduces the manufacturing costs. It is worth noting that the cathode chemistry of TPABr$_3$/Br$^-$ is a solid/liquid conversion, which effectively avoids the diffusion of liquid Br$_2$ into Zn anode to cause the crossover that is the side reaction in conventional Zn–Br$_2$ batteries. Therefore, the advanced anode-free Zn–Br$_2$ batteries eliminate the utilization of expensive ion exchange membrane and significantly reduce the overall battery costs.

$$\text{Cathode}: \text{TPABr} + 2\text{Br}^- \leftrightarrow \text{TPABr}_3 + 2e^- \qquad (1)$$

$$\text{Anode}: \text{Zn}^{2+} + 2e^- \leftrightarrow \text{Zn} \qquad (2)$$

$$\text{Overall}: \text{TPABr} + 2\text{Br}^- + \text{Zn}^{2+} \leftrightarrow \text{TPABr}_3 + \text{Zn} \qquad (3)$$

We explored the rate capability and long-term cycle stability of the Zn–Br$_2$ batteries with Sb@Cu anode to demonstrate their promises for practical energy storage applications. As shown in Fig. 5b, the anode-free Zn–Br$_2$ battery with an areal capacity of 10 mAh cm$^{-2}$ exhibits a CE of ~98% at 1 C and ~83% at a high rate of 10 C. The Zn–Br$_2$ battery also possesses a discharge voltage of 1.65 V at 1 C, and 1.1 V even at 8 C (Fig. 5c), demonstrating its attractive rate capacity. Excitingly, the Zn–Br$_2$ battery delivers an attractive energy density of ~274 Wh kg$^{-1}$ when taken active materials of both cathode and anode into account. Meanwhile, a practical energy density of 62 Wh kg$^{-1}$ is achieved on the basis of the entire Zn–Br$_2$ pouch cell, including all components of cathode, anode, separator, electrolyte, current collector, and packaging (Supplementary Figs. 14, 15). As

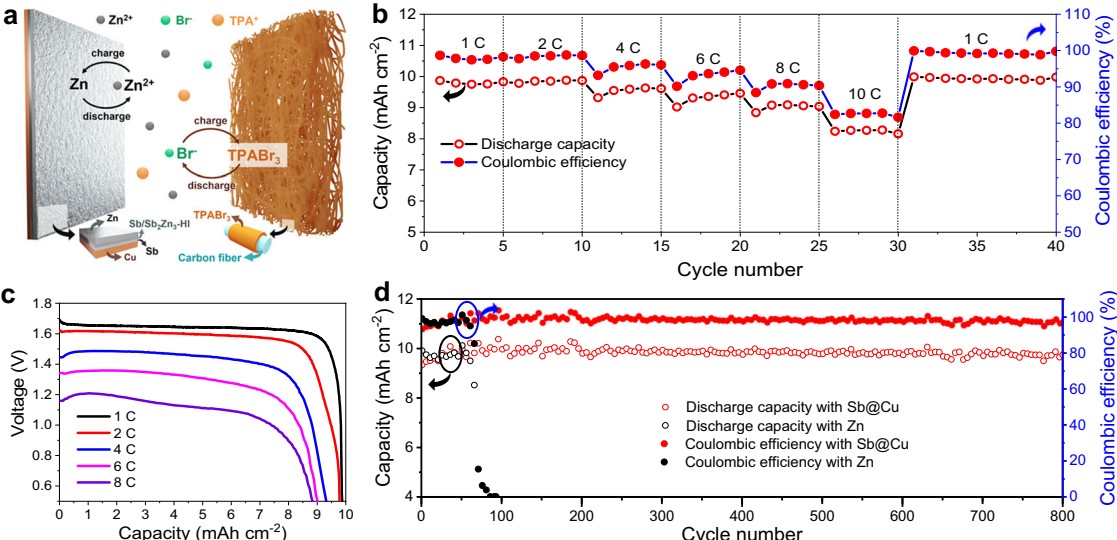

**Fig. 5 | Electrochemical performance of anode-free Zn–Br₂ battery with an areal capacity of 10 mAh cm⁻².** **a** A schematic diagram of the anode-free Zn–Br₂ battery. **b** Rate capacities in the current density range of 10 to 100 mA cm⁻². **c** Discharge curves at current densities from 10 to 80 mA cm⁻². **d** Long-term cycling performance at 10 mAh cm⁻² and 10 mA cm⁻². The electrochemical measurements of the Zn–Br₂ batteries were carried out at room temperature (25 °C) with a battery discharge cut-off voltage of 0.5 V.

illustrated in Fig. 5d, our Zn–Br₂ battery with Sb@Cu anode reveals negligible capacity decay over 800 cycles at an areal capacity of 10 mAh cm⁻² and a discharge current density of 10 mA cm⁻². In contrast, the battery with Zn foil anode can only be cycled steadily for about 50 cycles, followed by a rapid drop in efficiency and eventually complete failure in less than 100 cycles. This can be attributed to the uncontrollable Zn dendrite growth resulted short-circuiting of the battery. Even when the battery capacity is enlarged to 20 mAh cm⁻², a high average CE of 95% is still maintained after 170 cycles at a discharge current density of 20 mA cm⁻² (Supplementary Fig. 16). Our findings disclose that the exceptional cycle stability of the Zn–Br₂ battery is correlated to the highly reversible chemistries of Br⁻/TPABr₃ and Zn²⁺/Zn. It also enlightens our strategies for advancing the practical application of the next-generation Zn–Br₂ batteries.

## Electrochemical performance of Zn-free anode with ultrahigh capacities

To assess the potential for high energy batteries, the Sb@Cu anode with ultrahigh capacities have been further investigated. Fig. 6a shows the charge and discharge curves of the Zn|Sb@Cu half-cells with various capacities from 10 to 200 mAh cm⁻². Impressively, the Sb@Cu anode exhibits an ultrahigh areal capacity of 200 mAh cm⁻² with an overpotential of 112 mV and a high CE of 98.5% (Supplementary Fig. 17). A comparison between our Sb@Cu anode and the reported anode-free substrates for Zn plating was summarized by considering several key parameters, such as areal capacity, current density, and accumulated capacity (Fig. 6b). Our Sb@Cu substrates exhibit obvious advantages particularly in terms of the high areal capacity of 200 mAh cm⁻², which is about one to two orders of magnitude higher than the reported values[16,22,29,32–34,36–38,51,52]. In addition, the high current density of 200 mA cm⁻² and accumulated capacity of 5500 mAh cm⁻² are among the best reported values, suggesting the great promises of the Zn-free Sb@Cu anode for high energy Zn batteries. Optical microscope photographs were taken to show the Zn electrodeposition characteristics at a high areal capacity of 200 mAh cm⁻² (Fig. 6c). The Zn substrate has a rough surface with dendrites stacked around the edges. In contrast, the surface and edges of the Sb@Cu substrate exhibit dendrite-free morphology. Such a uniform Zn electrodeposition at ultrahigh areal capacities benefits from the regulating function of the Sb/Sb₂Zn₃-HI to

homogeneous Zn nucleation and continuous extension of the advantage to the galvanized Zn layer.

Even stronger evidence comes from the assembled Sb@Cu anode with ultrahigh areal capacity in an asymmetric anode-free Zn–Br₂ full cell. As illustrated in Fig. 6d, the battery is composed of 20 cm² CF cathode and 1 cm² Sb@Cu anode, where the designed capacity is 200 mAh. As a result, the battery exhibits a discharge voltage of ~1.5 V (Fig. 6e), a stable cycle approaching 40 times with an average CE of over 94% (Fig. 6f). The attractive electrochemical performance of the anode-free Zn–Br₂ battery indicates that the advantage of ultrahigh areal capacity of the Sb@Cu substrate is highly practical, further testifying its potential for commercialization.

## Scaled-up Zn–Br₂ batteries for practical energy storage applications

One of the most essential criteria for large-scale energy storage application of the battery is the electrochemical behaviors at enlarged capacities. In this regard, we scaled up the Zn–Br₂ battery to a capacity of 500 mAh through an alternating electrode stacking structure. As shown in Fig. 7a, the battery is composed of a carbon felt cathode, a glass fiber separator, a Zn-free Sb@Cu anode, and ZnBr₂ with TPABr electrolyte, in which two pairs of electrodes with individual area of 36 cm² (equivalent to an areal capacity of ~7 mAh cm⁻²) were applied. The stacked electrodes were assembled into a homemade plexiglas device with dimensions of 6.5 cm × 8 cm × 12 cm (Fig. 7b). Excitingly, the scaled-up Zn–Br₂ battery can be stably cycled for over 400 cycles and exhibited an average CE of 98.5% (Fig. 7c). Subsequently, we have further reduced the electrode thickness (1.5 mm for CF cathodes and ~0.03 mm for Sb@Cu anodes) and increased electrodes to three pairs. The designed battery exhibits a capacity of 1500 mAh (areal capacity of ~13.9 mAh cm⁻²). As shown in Fig. 7d, the individual battery exhibits a discharge capacity of 1393 mAh and a high discharge voltage of 1.5 V. Moreover, the batteries in different combinations still exhibit good electrochemical performance, including two parallel-connected battery with a discharge capacity of 2833 mAh and a discharge voltage of 1.6 V, two series-connected battery with a discharge capacity of 1389 mAh and a discharge voltage of 3.12 V, as well as two parallel plus two series-connected battery with a discharge capacity of 2740 mAh and a discharge voltage of 3.1 V. The acquired discharge energy of

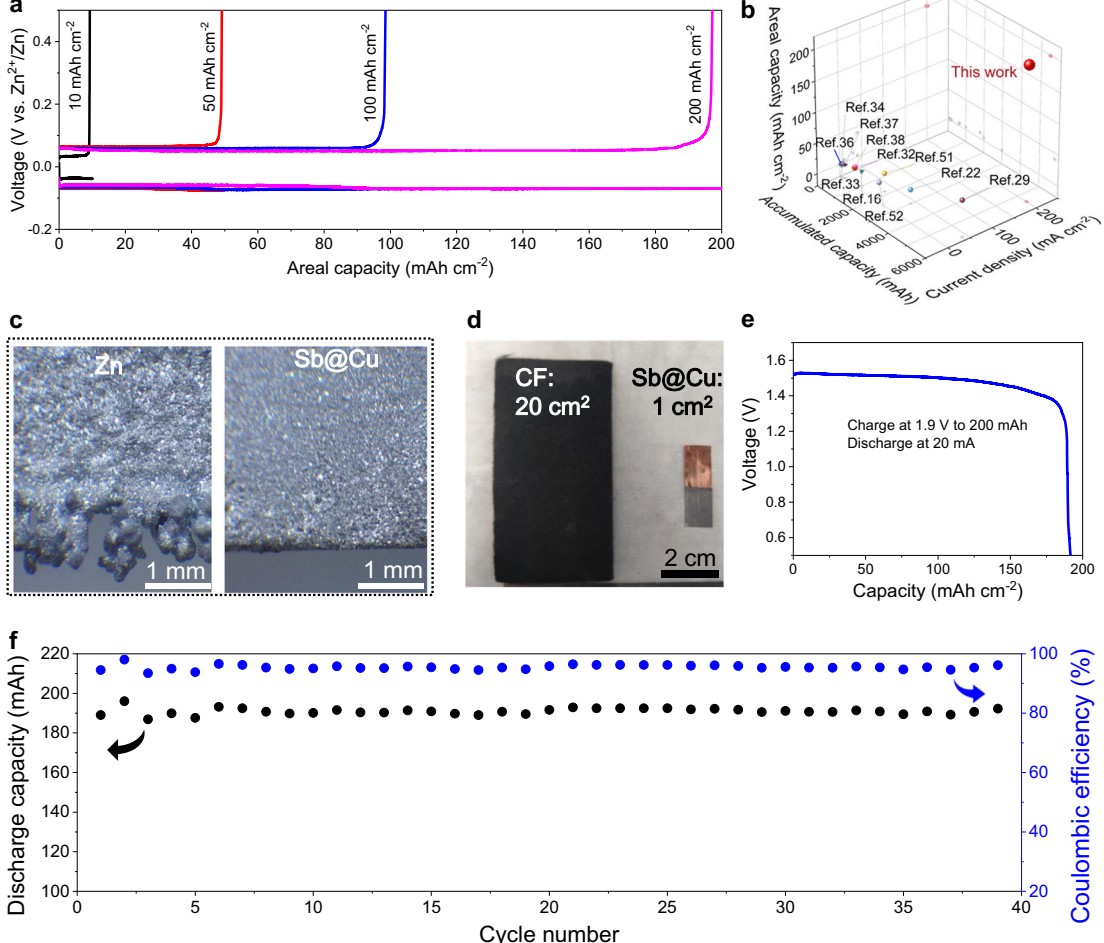

**Fig. 6 | Electrochemical performance of Zn electrodeposition at ultrahigh capacities on the Sb@Cu substrate. a** Charge and discharge curves of the Zn| Sb@Cu half-cells at different capacities with a constant current density of 20 mA cm$^{-2}$ and a cut-off voltage of 0.5 V vs. Zn$^{2+}$/Zn. **b** A summary of Zn-free electrodes for Zn plating/stripping in terms of areal capacity, current density, and accumulated capacity. **c** Digital photographs of Zn plating on Zn and Sb@Cu substrates at an areal capacity of 200 mAh cm$^{-2}$. Asymmetric anode-free Zn–Br$_2$ battery. **d** Digital photographs of the cathode and anode, where the battery has a carbon felt cathode of 20 cm$^2$ and a Sb@Cu anode of 1 cm$^2$ (CF represents carbon felt). **e** Discharge curve. **f** Cycling performance. The asymmetric anode-free Zn–Br$_2$ battery was charged at 1.9 V to 200 mAh and discharged at 20 mA to 0.5 V. The electrochemical measurements of the half-cells and Zn-Br$_2$ battery were carried out at room temperature (25 °C).

single cells in the battery module is slightly higher than that of the individual single cells, demonstrating the remarkable scalability and industrial application potentials of our Zn–Br$_2$ batteries.

The self-discharge performance is another important parameter to assess the practicality of batteries. Conventional Zn–Br$_2$ batteries suffer from severe self-discharge due to the crossover issue of the Zn anode by the diffusion of bromine. Thus, expensive ion exchange membranes are commonly used to mitigate this phenomenon, which significantly increase the battery cost[53,54]. In our work, low-cost glass fiber separator is constructed into a static Zn–Br$_2$ battery with a capacity of 1500 mAh to evaluate its self-discharge performance. As shown in Supplementary Fig. 18, when the fully charged battery is rest for 48 h, the open-circuit voltage only drops by 50 mV from the initial 1.813 V to 1.763 V. Meanwhile, it can still discharge a capacity of 1383 mAh, which is 95% that of the battery without the rest for 48 h, with a discharge voltage of 1.48 V. This can be ascribed to the remarkable TPABr$_3$/Br$^-$ solid/liquid redox chemistry, preventing bromine diffusion from contaminating the Zn anode, thus enabling the battery to suppress self-discharge even without the utilization of costly ion exchange separators.

The eye-catching performance of the scaled-up anode-free Zn–Br$_2$ battery has provoked our interest in demonstrating its potential towards renewable energy storage via the integration with photovoltaic cell panels. We connected four 1500 mAh Zn–Br$_2$ batteries in series into a module of ~9 Wh (6 V and 1.5 Ah), which is then charged by a photovoltaic cell panel (6 W and 9 V) for about 2 h during daytime. The solar-powered battery module can consistently light up a 10 W LED display in night (Fig. 7e). When discharging, the battery module still exhibited a discharge capacity of 942 mAh and a discharge voltage of 5.97 V after lighting up the LED display for 10 min (Supplementary Fig. 19). The successful integration of the Zn–Br$_2$ battery module with the photovoltaic cell panel illustrates their high adaptability and energy storage applications in future smart grids. It can be foreseen that further optimization of the battery components can unfurl a reinvigorated arena of next-generation anode-free Zn–Br$_2$ batteries with an attractive cost and remarkable energy density for future grid-scale energy storage applications.

## Discussion

We have developed a metal/metal alloy heterostructured interface to modulate Zn nucleation and stable Zn growth, which enabled dendrite-free Zn plating/stripping at an ultrahigh capacity of 200 mAh cm$^{-2}$. Benefiting from the Zn-free anode design, the Zn–Br$_2$ full battery achieved a competitive energy density of ~274 Wh kg$^{-1}$ with a practical

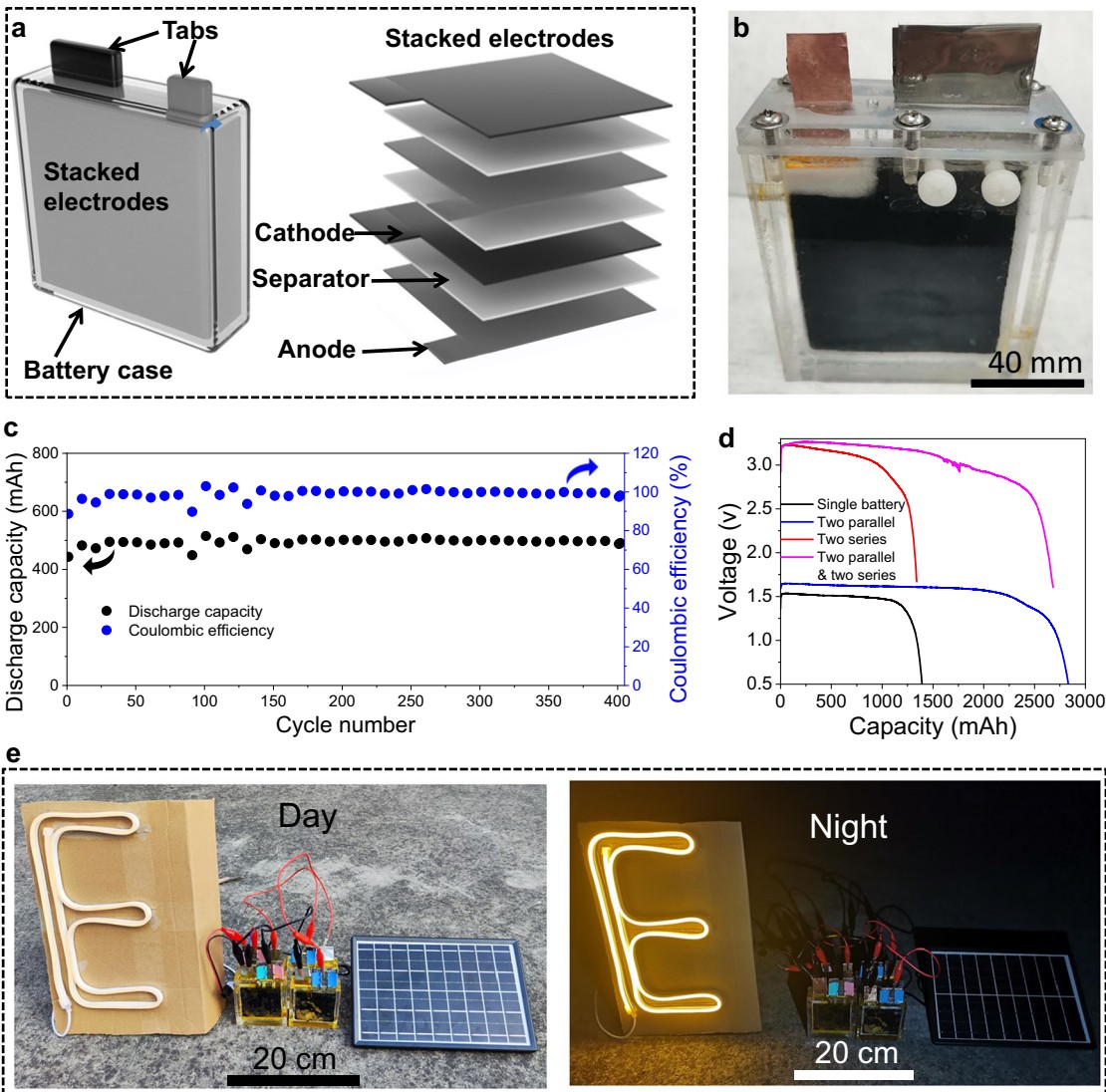

**Fig. 7 | Scaled-up Zn–Br₂ batteries for practical energy storage applications.** **a** Schematic of the scaled-up Zn–Br₂ battery structure. **b** A digital photograph of the assembled battery. **c** Cycling performance of the scaled-up battery with a capacity of 500 mAh. The battery was charged firstly at 250 mA to 1.95 V and then at 1.95 V to 500 mAh, and finally discharged at 250 mA to 0.5 V. **d** Discharge curves of the batteries with different connections. The individual battery was charged firstly at 300 mA to 2 V and then at 2 V to 1500 mAh, and finally discharged at 200 mA to designated voltage. **e** Solar powered battery energy storage system at day and night. The demonstrated system of solar powered energy storage is based on the Zn–Br₂ battery module as the energy storage device (four series-connected Zn–Br₂ batteries with a voltage of ~6 V and a capacity of 1500 mAh), photovoltaic cell panel as power source (rated at 9 W) and an LED display (rated at 10 W) serving as electrical load. The electrochemical measurements of the scale-up Zn–Br₂ batteries were carried out at room temperature (25 °C).

pouch cell energy density of 62 Wh kg⁻¹, which reflected its potential as energy storages devices. Furthermore, it can be cycled over 800 times without capacity decay, and the scaled-up battery with a capacity of 500 mAh also exhibited stable cycling over 400 times. The battery module with an energy of 9 Wh (6 V, 1.5 Ah) was successfully assembled using the enlarged 1.5 Ah Zn–Br₂ batteries, which can be integrated with the renewable energy source and demonstrated a discharge voltage of ~6 V to power LED displays. We believe that the anode-free batteries engineered by the metal/metal alloy heterostructured interfaces will enable a revolution towards their future energy storage applications.

## Methods
### Materials
Potassium bromide (KBr, ≥99%), bromine (Br₂, ≥99%), sulfuric acid (H₂SO₄, ≥ 95%) and copper foil (100 μm and 30 μm, 99.5%) were purchased from Sinopharm Group Co. Ltd. Zinc bromine (ZnBr₂, ≥99%),

tetrapropyl ammonium bromide (C₁₂H₂₈NBr, ≥98%) and antimony trichloride (SbCl₃, ≥99%) were procured from Shanghai Maclean Biochemical Technology Co. Ltd. Carbon felt (thickness: 3 mm and 1.5 mm) was adopted from Dalian Longtian Technology Co. Ltd. Zn foil (50 μm) was manufactured by Beijing Xinruichi Technology Co., Ltd. Ketjen black EC600JD was procured from Alibaba Group.

### Preparation of Sb@Cu current collector
To fabricate a clean and fluffy surface, the copper foil was first polished with fine sandpaper and afterwards cleansed with deionized water. The Sb plating on copper substrates was carried out using a conventional electroplating approach. It incorporates a two-electrode system with a copper working electrode and a graphite plate counter electrode. The electrolyte was comprised of 4 M H₂SO₄ and 0.03 M SbCl₃ dissolved in deionized water. During the electroplating process, the thickness of the Sb layers was tuned by adjusting the plating time (3–20 min) at a current density of 3 mA cm⁻².

## Battery fabrication

In 2 M $ZnBr_2$, half-cells were constructed in coin and beaker cell configurations, utilizing 80 μL electrolyte for coin cells. In half-cells, the substrates of Zn foils (50 μm), Cu foil (100 μm) and Sb@Cu (~100 μm) were acted as the working electrodes, and Zn foil as the counter electrode. The tests of Zn full batteries were conducted in a home-made plexiglas device with 3 mL electrolyte, as shown in Supplementary Fig. 13. In the Zn–$Br_2$ full cell, carbon felt (thickness of 3 mm) and the Sb@Cu (thickness of ~100 μm) with an active area of 1 $cm^2$ was used as the cathode and anode, respectively, where the electrolyte consists of 0.5 M $ZnBr_2$ and 0.25 M TPABr.

The scaled-up Zn–$Br_2$ battery was integrated in a plexiglas device with the dimensions of 65 mm × 12 mm × 80 mm (Fig. 7a). The electrolyte was a mixture of 0.5 M $ZnBr_2$ and 0.25 M TPABr in aqueous solution in an amount of 55 ml. The electrodes (thickness of carbon felt is 3 mm and that of Sb@Cu is ~0.1 mm) were assembled via lamination into two pairs with an active area of 36 $cm^2$ and a designed capacity of 500 mAh (Fig. 7b). In addition, a glass fiber separator with a thickness of 1 mm is placed between the carbon felt and the Sb@Cu terminals to prevent the battery from short-circuiting. In the further enlarged battery with a capacity of 1500 mAh, the design has been increased to three pairs of electrodes. The carbon felt (thickness of 1.5 mm) and Sb@Cu (thickness of ~30 μm) are functionally operated as the cathode and anode current collectors, respectively. The Sb@Cu was pre-deposited with ~15 mAh $cm^{-2}$ Zn. It is noteworthy that to inflate carbon felt cathode's areal capacity in the 1500 mAh battery, we embellished positive materials upon carbon felt with a mass loading of 69 mg $cm^{-2}$ (the mass percentage of active material $TPABr_3$ is about 87%) via titration. To prepare the electrode slurry, $TPABr_3$, ketjen black-EC600JD and PVDF with a mass ratio of 100:10:5 were mixed in N-methylpyrrolidone (NMP) solvent to craft a homogenized slurry. The as-prepared slurry was rationed evenly upon carbon felt and dried in an electric oven at 80 °C for 6 h, and the operation was repeated 3 times to reach the rated mass loading.

## Characterizations

Morphological analysis of anode microstructure was carried out using optical microscope (MICV), scanning electron microscope (SEM, JEOL-6700F) and transmission electron microscope (TEM, JEOL JEM-2100F). For elemental analysis, X-ray diffraction (XRD, Smart Lab, Japan) with Cu $K_\alpha$, and X-ray photoelectron spectroscopy (XPS, Thermo ESCALAB 250Xi) with a monochromatic Al $K_\alpha$ source of 1486.6 eV were employed.

## Electrochemical measurements

Tafel plots and cyclic voltammetry (CV) were performed in a typical three-electrode system with the EC-Lab electrochemical workstation (Biologic VMP3, France), where the substrates of Zn foil, Cu foil, Sb@Cu and 0.2 mAh Zn plated Sb@Cu were acted as the working electrodes, graphite rod as the counter electrode and Ag/AgCl as the reference electrode. The Tafel and CV plots were tested in 2 M KBr, where the scan rate of the Tafel plots was 5 mV $s^{-1}$ with a potential of ± 0.25 V vs. open-circuit potential, and that of the CV plots was 5 mV $s^{-1}$ in a voltage range of −1.2 ~ −2.5 V vs. Ag/AgCl. The charge/discharge measurements of the half and full batteries were recorded via battery test systems (LandHe, Wuhan, China; NEWARE, Shenzhen, China). Specifically, the half-cell tests were executed by applying a constant current, where the cut-off voltage for the Zn|Sb@Cu half-cells was 0.1 V for coin cells and 0.5 V for beaker cells. The Zn–$Br_2$ batteries were charged to a certain capacity (10 mAh and 20 mAh) at 1.9 V and discharged to 0.5 V vs. $Zn^{2+}$/Zn at 10–80 mA $cm^{-2}$. The scaled-up Zn–$Br_2$ battery was firstly charged under constant currents to certain voltages and then charged at a constant voltage to the quantified capacity, thereafter discharged at a constant current. All of the electrochemical measurements were conducted in air at room temperature of 25 °C.

## Theoretical calculation and simulation details

The first principles calculations were carried out with the standard DFT using Vienna ab initio Simulation Package (5.4.4 VASP) within the generalized gradient approximation (GGA) as formulated by the Perdew−Burke−Ernzerhof (PBE) functional. The Zn (100) and Sb (104) surfaces with six atoms were modeled for the adsorption calculation with a 20 Å vacuum space to avoid the interaction from nearby layers, where the layers 4 to 6 are the central layers and the layers 1 to 3 are the surface layers. The final set of energies for all calculations was computed with a cut-off energy of 520 eV. The convergence criteria for energy were set to be $10^{-5}$ eV, and the residual forces on each atom became smaller than 0.02 eV $Å^{-1}$. The Brillouin zone integration was performed with 4 × 4 × 1 and 2 × 3 × 1 Γ-centered Monkhorst-Pack k-point meshes for Zn (100) and Sb (104) in the geometry optimization calculations. To further confirm the adsorption effect of the Sb substrate after the Zn/Sb alloy, successive adsorptions were continued until the 21st atom was collected.

The electrochemistry and material transport infusion of Zn electrodeposition was simulated using COMSOL Multiphysics via coupling the "Second current distribution" with "Transport of Diluted Species" fields. The electrolyte concentration is preset to 2 M of $ZnBr_2$. The linear Bulter−Volmer equation (Eq. 4) was utilized to compute the electrode reaction kinetics. The $i_{loc}$ denotes the local current densities over the electrode and electrolyte interface, whereas $i_O$ is the exchange current density. Similarly, $\alpha_a$ and $\alpha_c$ are the anodic and cathodic transfer coefficients, and the $F/RT$ is the Nernst parameter.

$$i_{loc} = i_0 \left( \frac{(\alpha_a + \alpha_c)F}{RT} \right) \eta \qquad (4)$$

Furthermore, the Nernst−Einstein relation (Eq. 5) was used to compute ion diffusion, where $D$ reflects diffusion coefficient, $k$ marks Boltzmann constant, and $T$ symbolizes temperature. To acquire the current density distribution and Zn ion flow, we configured the electrodeposition parameters of Zn to 700 s under transient conditions and generated the data at an interval of 5 s. The physical parameters were shown in Supplementary Table 3.

$$D = \mu k T \qquad (5)$$

## Calculation of energy density of the anode-free Zn batteries

The energy density of Zn batteries was reckoned to assess their ultimate potential for future large-scale energy storage applications. The calculation of energy density (ED) is based on positive and negative active materials (Eq. 6). The anode-free Zn–$Br_2$ battery with a capacity of 10 mAh portrays a discharge energy (E) of 16.17 mWh, where the initial plating mass of the $TPABr_3$ cathode is 0.0468 g and 0.01219 g for the Zn anode, showing a total mass (m) of 0.05899 g. Thus, the Zn–$Br_2$ battery displays an energy density of 274.1 Wh $kg^{-1}$ with respect to the active materials of cathode and anode only. The practical energy density of the Zn–$Br_2$ pouch cell was calculated based on all components, including cathode, anode, separator, electrolyte, current collector, and cell packaging, rendering a promising value of 62 Wh $kg^{-1}$.

$$ED = \frac{E}{m} \qquad (6)$$

## Data availability

All data generated in this study are provided in the Source Data file and its Supplementary Information. Source data are provided with this paper.

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

## Acknowledgements

W.C. acknowledges the funds from USTC (Grant # KY2060000150) and the Fundamental Research Funds for the Central Universities (WK2060000040). X.Z acknowledges the Fundamental Research Funds for the Central Universities (WK5290000003). We thank the support from USTC Center for Micro and Nanoscale Research and Fabrication, and USTC Supercomputer Center.

## Author contributions

X.Z. and W.C. conceived the idea. X.Z. and R.L. designed the cells and conducted the electrochemical measurements. X.Z. conducted SEM, XRD, XPS, and TEM characterization. J.S. helped with XPS and XRD analysis. Z.L. conducted the DFT calculations. X.Z. conducted the COMSOL simulations. M.S and J.K conducted the local electrochemical measurements. W.C. supervised the project. X.Z. and W.C. contributed to writing the manuscript. X.Z, Z.L., J.S., R.L., K.X., M.S., J.K., Y.Y., S.L., T.A., T.J., N.C., M.W., Y.X., M.C., Z.Z., Q.P., Y.M., K.Z., W.W., and W.C. discussed the results.

## Competing interests

The authors declare no competing interests.
