## [Peer Review File · Nature Communications]

Constructing robust heterostructured interface for anode-free zinc batteries with ultrahigh capacitiesREVIEWER COMMENTS

Reviewer #1 (Remarks to the Author):

The manuscript entitled "Constructing robust heterostructured interface for anode-free zinc batteries with ultrahigh capacities" demonstrate an antimony/antimony-zinc alloy heterostructured interface (Sb/Sb₂Zn₃-HI) to regulate Zn nucleation and growth. Benefiting from the stronger adsorption and homogeneous electric field distribution of the Sb/Sb₂Zn₃-HI in Zn plating, the Zn anode enables an ultrahigh areal capacity of 200 mAh cm⁻² with an overpotential of 112 mV and a Coulombic efficiency of 98.5%. The authors also coupled Sb/Sb₂Zn₃-HI@Cu anode with bromine cathode to fabricate an anode-free Zn-Br₂ battery, which shows an attractive energy density of 274 Wh kg⁻¹ and over 400 cycles in 500 mAh scaled-up configuration. The authors further integrated the Zn-Br₂ battery module with a photovoltaic panel to demonstrate the practical renewable energy storage capabilities.

I agree that antimony/antimony-zinc alloy coating on Cu substrate is an efficient way to eliminate Zn dendrite growth during Zn electrodeposition. However, I noticed that quite similar strategies employing antimony/antimony-zinc alloy as an interface to suppress Zn dendrites on Zn anode in aqueous batteries had been reported in earlier papers (Advanced Science, 2022, 9(6): 2104866; Energy Storage Materials, 2021, 41: 343-353; Journal of Colloid and Interface Science, 2020, 579: 823-831). In addition, the Zn suppression mechanism demonstrated in this paper is quite similar to the corresponding mechanism reported in previous literatures, which is not novel at all.

Moreover, the authors failed to show me a clear picture on the Zn²⁺ diffusion behavior inside the heterostructured interface (Sb/Sb₂Zn₃-HI) during electrodeposition. As a result, I think both the novelty and quality of this work does not meet the requirement of Nature Communications, and rejection is suggested.

Below I list the critical points the authors have to address before submitted to another journal:

- 1) The authors claimed that Sb/Sb₂Zn₃-HI shows a stronger adsorption for Zn atoms based on theoretical calculations. However, more electrochemical/spectrum characterizations should be provided to support this conclusion.
- 2) Why does Sb/Sb₂Zn₃-HI forms homogeneous electric field distribution, and how does this electric field enable uniform Zn deposition? The authors should give us more characterizations on these points to help us more clearly understand the Zn²⁺ diffusion process inside Sb/Sb₂Zn₃-HI.
- 3) There are no equivalent circuits in the Nyquist plot of Figure S6. All the EIS data in the manuscript should be fitted and the corresponding equivalent circuit and fitting parameters should be provided.
- 4) The high energy density of ~274 Wh kg⁻¹ in this study is based on the mass of active materials. It is quite misleading. The total mass of the devices (including substrate, shell and separator) should take into account for the evaluation of energy densities, which is more meaningful for practical applications.
- 5) Zn is highly reactive and forms SEI layers, and the competing surface reactions lead to complex "mossy" growth below the diffusion limited current (steady state) or before "Sand's time" (for galvanostatic over-limiting current). The following paper visually and electrochemically demonstrates the transition between mossy and dendritic growth of lithium at "Sand's capacity": Bai, Peng, et al. "Transition of lithium growth mechanisms in liquid electrolytes." Energy & Environmental Science 9.10 (2016): 3221-3229. The authors must check what is Sand's capacity for their battery systems and explain whether their experiments are carried out in the dendritic growth regime. Bai showed that many papers on lithium full cells that claim to "solve the dendrite problem" in fact are operating below Sand's

capacity where dendrites would not occur anyway. This means the current was small or the time was short (too small capacity).

6) The authors are encouraged to evaluate the dendrite control effect of Sb/Sb₂Zn₃-HI in practical cell conditions, e.g. lean electrolyte or low N/P ratio, to validate the feasibility of this strategy in industrial level batteries.

Reviewer #2 (Remarks to the Author):

This article reported an Sb/Sb₂Zn₃ heterostructured interface to regulate Zn nucleation and growth in Zn batteries. Experimental and theoretical investigations are implemented to unveil the Zn deposition mechanism. This finding is interesting, and this article deserves publication in Nature communication after answering the following questions:

Q1: In line 184, the authors claimed, "This suggests the uniform Zn plating can be well maintained even for thick Zn deposits." It is hard to get such a firm conclusion from the DFT results. In the Figure 2a, we could only get the adsorption energy versus the Zn number without the geometric information.

Q2: In the Figure 2c, it seems that there are Sb atoms around adsorbed Zn atoms, which explains the claim in line 187 "deposited Zn atoms on the Sb surface are more delocalized than that on the Zn surface." Why is the adsorption configuration different between Zn(001) and Sb(104)?

Q3: For the phase field analysis, please add the scale in Figure 2d. Also, please tabulate the physical parameters.

Q4: In the Figure S7, why the Zn flux in Sb/Sb₂Zn₃ base is two orders less than in pure Zn base?

Reviewer #3 (Remarks to the Author):

In this manuscript, the authors designed a robust antimony/antimony-zinc alloy heterostructured interface (Sb/Sb₂Zn₃-HI) to stabilize Zn nucleation and electrodeposition. Benefiting from the effects of stronger adsorption and homogeneous electric field distribution of the Sb/Sb₂Zn₃-HI in Zn plating, the dendrite-free Zn electrodeposition was achieved even at an ultra-high areal capacity of 200 mAh cm⁻². The realization of such a record high areal capacity of Zn plating is truly a breakthrough, offering the Zn-free Sb@Cu anodes a greater practical value. I have never seen from previous reports on such a high areal capacity of Zn anode. The authors then conducted various materials characterization, DFT calculation and simulation to reveal the mechanism of the anode-free Zn electrodeposition by the HI design. Furthermore, the anode-free Zn-Br₂ battery based on the Sb/Sb₂Zn₃-HI@Cu anode exhibited a high energy density and good cycling stability at an areal capacity of 10 mAh cm⁻². The authors further scaled the Zn-Br₂ battery up to Ah level, which displayed high energy density for the practical battery storage module by charging from photovoltaic cells. Overall, the anode material reported in this manuscript is very innovative and the achieved Ah-level batteries offer promises for the practical application of Zn batteries. This work provides an important advancement for the future design of Zn anode. Therefore, I strongly support this

manuscript to be accepted after some minor revisions. The specific comments are as follows.

1. In the introduction, the authors mentioned that "Using well-engineered zincophilic substrates as anode current collectors can effectively inhibit the formation of Zn dendrites, while making them feasible to achieve anode-free Zn battery design". The zincophilic substrates are a broad concept, the authors need to list specific materials or categories to further inform the reader about the zincophilic materials.

2. In the "Design of 2D Heterostructured Interface" section, the authors mentioned that "Cu was selected as the substrate for the Zn-free anode to build up a durable Sb@Cu electrode (Supplementary Fig. 1)". Here the authors should add the explanation of the mechanism on why Sb and Cu can form a durable interfacial layer.

3. The rate capacities shown in Figure 4c do not match with the description. The authors need to further check and update this part.

4. The authors performed a Zn-Br₂ battery cycle performance test using only Sb@Cu anodes. Additional Zn-Br₂ battery cycle performance tests using Zn foil as the anode are needed for better comparison.

5. The electrolyte will affect the electrochemical performance of energy storage systems. You can do some corresponding research to improve the stability of your battery, at least cite some relevant papers to give a potential promotion method, just example: Nat. Sustain., 2022, 5, 225-234; DOI: 10.1002/anie.202208291; National Science Review, nwac134.

6. The authors carried out a scale-up study of Zn-Br₂ battery, showing good Coulombic efficiency. But its cycling stability is still much worse than that of small Zn-Br₂ battery (800 cycles), what are the key issues for the enlarged Zn-Br₂ battery?

7. The authors reported an attractive energy density of the Zn-Br₂ battery (about 274 Wh/kg), which was based on the active materials of anode and cathode. Did the authors evaluate the energy density of the entire scaled-up battery? This should include the active materials, current collectors, electrolyte, etc. It will help to demonstrate the practical application of the battery.

Response to Reviewers' Comments

We thank the reviewers for their valuable comments, which we find very constructive and valuable. We are delighted that all three reviewers have objectively assessed our work. We have addressed all the comments from the reviewers to the largest possible extent. For ease of reference, the reviewers' comments and suggestions are reproduced in **blue**, our response is in **black**, and the resulting changes to the manuscript are in **red**.

Reviewer #1 (Remarks to the Author):

General comment: The manuscript entitled "Constructing robust heterostructured interface for anode-free zinc batteries with ultrahigh capacities" demonstrate an antimony/antimony-zinc alloy heterostructured interface (Sb/Sb₂Zn₃-HI) to regulate Zn nucleation and growth. Benefiting from the stronger adsorption and homogeneous electric field distribution of the Sb/Sb₂Zn₃-HI in Zn plating, the Zn anode enables an ultrahigh areal capacity of 200 mAh cm⁻² with an overpotential of 112 mV and a Coulombic efficiency of 98.5%. The authors also coupled Sb/Sb₂Zn₃-HI@Cu anode with bromine cathode to fabricate an anode-free Zn-Br₂ battery, which shows an attractive energy density of 274 Wh kg⁻¹ and over 400 cycles in 500 mAh scaled-up configuration. The authors further integrated the Zn-Br₂ battery module with a photovoltaic panel to demonstrate the practical renewable energy storage capabilities.

I agree that antimony/antimony-zinc alloy coating on Cu substrate is an efficient way to eliminate Zn dendrite growth during Zn electrodeposition. However, I noticed that quite similar strategies employing antimony/antimony-zinc alloy as an interface to suppress Zn dendrites on Zn anode in aqueous batteries had been reported in earlier papers (Advanced Science, 2022, 9(6): 2104866; Energy Storage Materials, 2021, 41: 343-353; Journal of Colloid and Interface Science, 2020, 579: 823-831). In addition, the Zn suppression mechanism demonstrated in this paper is quite similar to the corresponding mechanism reported in previous literatures, which is not novel at all.

Moreover, the authors failed to show me a clear picture on the Zn²⁺ diffusion behavior inside the heterostructured interface (Sb/Sb₂Zn₃-HI) during electrodeposition. As a result, I think both the novelty and quality of this work does not meet the requirement of Nature Communications, and

rejection is suggested.

Our response: We would like to thank the reviewer for his/her thorough reading and insightful feedback. We are glad to have the reviewer's acknowledgement on the major achievement of our study. Meanwhile, the reviewer's critical comments will help us to further elaborate on the mechanisms of Zn electrodeposition and to improve the quality of our manuscript. We have tried our best to revise our manuscript and made substantial revisions as per the comments of the reviewer. All changes have been highlighted in red color in the revised manuscript and supporting information. We hope that the revised manuscript is suitable for publication in Nature Communications.

We thank the reviewer for mentioning a few previous papers on the involvement of antimony/antimony-zinc alloy as an interface to suppress Zn dendrites on Zn anode in aqueous batteries. However, we disagree with the reviewer's comments that our work is similar to the previous ones and our work shows no novelty at all. In fact, the working mechanism and the function of our designed antimony/antimony-zinc alloy heterostructured interface (Sb/Sb₂Zn₃-HI) are completely different from the previous work. First, we would like to interpret the differences between our work and the mentioned three papers. For the paper of Advanced Science, 2022, 9(6): 2104866, the authors reported an antimony (Sb) protective layer on Zn foil surface to prevent hydrogen and dendrite formation in Zn-ion battery. The mechanism of stable Zn electrodeposition is attributed to the abundant Zn nucleation sites and uniform electric field on the Zn anode surface, which does not involve the alloying of Sb and Zn and the formation of a stable Sb/Sb₂Zn₃-HI. Furthermore, the anode for this work is still Zn foil, and thus the utilization of the active Zn is relatively low. The coating protection strategy shown in this paper has a completely different mechanism from the anode-free strategy proposed in our manuscript. For the paper of Energy Storage Materials, 2021, 41: 343-353, the authors used zincophilic antimony engineered MXene@Sb enables a chemistry of alloying/dealloying of Zn with Sb to inhibit Zn dendrite formation and stable Zn batteries. This mechanism of stable Zn electrodeposition is the alloying of zincophilic Sb with Zn and does not involve the interfacial layer of metal/metal alloy with Zn, which is different from our heterostructured interface mechanism. In addition, the energy storage mechanism of Zn anode is Zn alloying and dealloying with Sb, which is not the same as Zn plating/stripping mechanism reported in our manuscript. Further, the authors reported that the maximum areal capacity of the Zn anode in the half-cell is only 2 mAh cm⁻² for 300 cycles, while

our Sb@Cu stabilized the Zn plating/stripping at 10 mAh cm⁻² for 550 h and obtained an ultra-high areal capacity of 200 mAh cm⁻². For the paper of Journal of Colloid and Interface Science, 2020, 579: 823-831, the authors developed a carbon-framed ZnO/Sb anode, and the main function of the Sb in the electrode is to suppress the corrosion and improve the interparticle conductivity. The deposition/dissolution of Zn was not involved in this work, and the mechanism has no relationship with the heterostructure interfacial layer in our work. In brief, although we all used Sb in the anode, the design of our Sb/Sb₂Zn₃-HI, its working mechanism and function are completely different from all the previous work. In addition, we summarized in Table R1 a comparison of the key electrochemical properties of the Zn batteries in our manuscript with the above mentioned three papers, which further illustrates the differences and the remarkable effect of our designed Sb/Sb₂Zn₃-HI on stabilizing Zn electrodeposition.

Table R1. A comparison of the key electrochemical performance of our manuscript with literature.

Papers	Current density (mA cm ⁻²)	Maximum areal capacity (mAh cm ⁻²)	Retention/Cycles or time (Areal capacity, discharge current density)	Accumulated capacity	Full battery maximum energy
This work	200	200	~98%/550 h (10 mAh cm ⁻² , 20 mA cm ⁻²)	5.5 Ah	9 Wh
Advanced Science	10	1	100%/1000 h (1 mAh cm ⁻² , 3 mA cm ⁻²)	1.5 Ah	NA
Energy Storage Material	10	2	~90%/300 (2 mAh cm ⁻² , 10 mA cm ⁻²)	0.6 Ah	NA
Journal of Colloid and Interface Science	30	613 mAh g ⁻¹ Zn	92%/2000 (~135 mAh g ⁻¹ for Ni-Zn)	NA	NA

We then would like to highlight the innovations of our manuscript. (1) We designed a completely independent Sb@Cu current collector, which not only adopts the anode-free design concept to reasonably control the N/P ratio of the Zn-Br₂ battery to be close to 1 but also maintains the integrity of the battery throughout the long-term cycling. (2) We applied industrial mature and

scalable electroplating method to prepare the Zn-free Sb@Cu anode, which is ideal for practical commercial Zn battery applications. (3) The Sb@Cu anode forms a robust antimony/antimony-Zn alloy heterostructure interface (Sb/Sb₂Zn₃-HI), which can effectively inhibit the formation of Zn dendrites even at a high areal capacity of 200 mAh cm⁻², a value that is about 1 to 2 orders of magnitude higher than that reported in the available literature. (4) The Zn-Br₂ cell displayed a high energy density of 274 Wh kg⁻¹ when taken into account both cathode and anode active materials. We have also conducted a systematic study of scaled-up Zn-Br₂ batteries to obtain a competitive energy density of 62 Wh kg⁻¹ and demonstrate their practicality for integrating renewable energy sources.

Regarding the Zn²⁺ diffusion behavior of our Sb/Sb₂Zn₃-HI, we would like to point out that we designed an anode-free current collector for Sb@Cu in this manuscript, where the initially plated Zn can be spontaneously alloyed with Sb to form the Sb/Sb₂Zn₃-HI and the Zn plating occurs on the surface rather than inside the Sb/Sb₂Zn₃-HI. In order to clarify the diffusion behavior of Zn²⁺ on the Sb/Sb₂Zn₃-HI, we further conducted additional experiments of local electrochemical tests (including impedance (Figure R3) and current density (Figure R5) in different regions of the sample surface), in-situ morphology observation of the Zn electrodeposition on the surfaces of Zn foil and Sb/Sb₂Zn₃-HI (Figure R6), as well as the calculation of the diffusion energy barrier of Zn atoms on the surface of Sb/Sb₂Zn₃-HI using DFT (Figure R7, R8 and R9). These results are reflected in the following specific comments.

Below I list the critical points the authors have to address before submitted to another journal:

Comment 1: The authors claimed that Sb/Sb₂Zn₃-HI shows a stronger adsorption for Zn atoms based on theoretical calculations. However, more electrochemical/spectrum characterizations should be provided to support this conclusion.

Our response: Thank the reviewer so much for the helpful comment. We have conducted additional experiments including the analysis of electrochemical impedance spectroscopy and the local electrochemical impedance spectroscopy tests to elaborate the stronger adsorption of the Sb/Sb₂Zn₃-HI for Zn atoms.

The impedance analysis of the substrate in the ZnBr₂ will provide a good verification of the interfacial adsorption energy. Therefore, we made a further detailed analysis of the impedance of

different substrates (Zn foil, Cu foil, Sb@Cu) in 2 M ZnBr₂ electrolyte. The equivalent circuit is shown in Figure R1, where the R_s is the ohmic resistance of solution and electrodes, R_{ct} is the charge transfer resistance, C_F is the double-layer capacitance, and Z_W is the Warburg impedance. We also included the fitted curves as shown in Figure R2. It can be seen from the fitting results in Table R2 that the Sb@Cu substrate displays an ohmic resistance of 1.95 Ω , which is lower than the 2.56 Ω of Zn foil. The smaller ohmic impedance exhibited by the Sb@Cu electrode indicates less energy loss on the electrochemical measurement system, and is thus favorable for the electrochemical reaction. Meanwhile, Sb@Cu also exhibits a lower charge transfer resistance of 2.62 Ω than the 19.78 Ω of the Zn foil, indicating that the reduction reaction of Zn²⁺ ions is more inclined to occur on the surface of Sb/Sb₂Zn₃-HI, which further verifies the stronger adsorption energy of Sb/Sb₂Zn₃-HI for Zn atoms. The updated contents were highlighted in the revised manuscript and also specified as follows:

(Page 8) *As illustrated in Supplementary Fig. 6, 7 and Table 2. The Sb@Cu substrate displays an ohmic resistance of 1.91 Ω , which is lower than 2.56 Ω of the Zn foil. Meanwhile, it also exhibits a lower charge transfer resistance of 2.62 Ω than 19.78 Ω of the Zn foil, indicating that the low adsorption energy on the Sb surface can effectively accelerate the charge transfer.*

Figure R1 An equivalent circuit is used to simulate the resistances of different substrates, where R_s is the ohmic resistance, R_{ct} is the charge transfer resistance, C_F is the double-layer capacitance, and Z_W is the Warburg impedance.

Figure R2. Electrochemical impedance spectroscopy of the Zn, Cu and Sb@Cu substrates and the corresponding fitting curves in 2 M ZnBr₂.

Table R2. The fitting results of EIS plots of all the samples in 2 M ZnBr₂.

Samples	R _s	R _{ct}	C _F
Sb@Cu	1.91	2.62	0.017
Cu	5.82	4.99	5.55e-5
Zn	2.56	19.78	3.06e-5

To further analyze the impedance information of the substrate surface, we scanned the impedance of a small area (1 mm * 1 mm) of the Zn foil and Sb@Cu substrate surface using a local electrochemical workstation (VersaSCAN, USA), where the step size was set to 50 μm and the impedance values were obtained for 400 points. As shown in Figure R3a, the Sb/Sb₂Zn₃-HI@Cu substrate displays lower impedance values of less than 1*e7, and the overall morphology is in a relatively flat state. In contrast, the Zn foil surface exhibits a larger overall impedance value than the Sb/Sb₂Zn₃-HI@Cu surface and shows more protruding points, indicating a non-uniform

distribution of impedance values on the Zn foil surface (Figure R3b). The cloud shape shows that the electric field on the surface of Sb/Sb₂Zn₃-HI@Cu is more uniform, and the numerical value shows that the Zn²⁺ ions are more active on the surface of Sb/Sb₂Zn₃-HI@Cu, which further support the conclusion of the stronger adsorption energy of the Sb/Sb₂Zn₃-HI for Zn atoms.

Figure R3. Local electrochemical impedance spectroscopy (LEIS) of the substrates. (a) Sb/Sb₂Zn₃-HI @Cu. (b) Zn foil.

Comment 2: Why does Sb/Sb₂Zn₃-HI forms homogeneous electric field distribution, and how does this electric field enable uniform Zn deposition? The authors should give us more characterizations on these points to help us more clearly understand the Zn²⁺ diffusion process inside Sb/Sb₂Zn₃-HI.

Our response: Thanks for the reviewer’s helpful comments. We would like to respond to the comments in the following three sections of “Why does Sb/Sb₂Zn₃-HI form homogeneous electric field distribution?”, “How does the electric field enable uniform Zn deposition?”, and “The diffusion behaviors of Zn²⁺ on the Sb/Sb₂Zn₃-HI”.

Why does Sb/Sb₂Zn₃-HI form homogeneous electric field distribution?

The initially plated Zn can spontaneously alloy with Sb to form the Sb/Sb₂Zn₃-HI, and the subsequent Zn plating occurs on the surface of the antimony/antimony-Zn alloy heterostructure interface rather than inside the Sb/Sb₂Zn₃-HI. As illustrated in the phase diagram of Figure R4, antimony is a Zn-soluble material, and thus Zn nucleation on Sb shows a low potential barrier. The formation of alloys between Zn and Sb is spontaneous during the electrochemical reduction of Zn, as a result of the negative Gibbs free energy of the formation of Zn and Sb alloys (*ACS Energy Lett.* **6**, 404-412, doi:10.1021/acsenerylett.0c02343 (2021)). Since Zn²⁺ ions are susceptible to being

reduced and then alloyed in all regions of the Sb surface, the overall distribution of the current is uniform at the entire interface. The DFT results show that after the formation of a robust Sb/Sb₂Zn₃-HI, the Zn/Sb alloy still maintains a strong tendency to adsorb the subsequently deposited Zn atoms (Figure 2b,c). Meanwhile the COMSOL simulation directly shows the uniform current density distribution on the Sb/Sb₂Zn₃-HI (Figure 2e).

To further understand the function of the Sb/Sb₂Zn₃-HI surface on uniform current density distribution and Zn²⁺ diffusion, more direct evidence is shown in the electrochemical properties of the local area scan of the substrates. We carried out additional experiment of LEIS as illustrated in Figure R3, the flatter and lower impedance values indicate a trend towards uniformity of current density on the surface of the Sb/Sb₂Zn₃-HI@Cu substrate and a faster diffusion trend of Zn²⁺. We further conducted new measurement of the scanning vibration electrode technique (SVET) to observe the current density distribution in localized regions on the surface of the Sb@Cu electrode. As can be seen from Figure R5, the average current density values on the surface of Sb/Sb₂Zn₃-HI is about 100 mA cm⁻², which is higher than that of Zn foil surface of ~30 mA cm⁻², indicating that the diffusion activity of Zn²⁺ in the Sb/Sb₂Zn₃-HI surface is stronger than the bare Zn surface. The current density on the Sb/Sb₂Zn₃-HI surface is much flatter in the cloud plot, which further indicates the uniformity of its surface electric field distribution.

Figure R4. Phase diagram of metal Zn and Sb. Adopted from FactSage.

Figure R5. Local current density of the substrates. (a) Sb/Sb₂Zn₃-HI@Cu. (b) Zn foil.

How does the electric field enable uniform Zn deposition?

Originally, Zn²⁺ ions are in favor to nucleate on the preferential adsorption sites (where the energy barrier is lower), which is determined by the uneven energy barrier of the imperfect surface of the Zn foil current collector (*Nat. Energy* **5**, 743-749, doi:10.1038/s41560-020-0674-x (2020)). The subsequent Zn electrodeposition prefers to occur at previous nucleation tips due to their lower energy barriers than the Zn²⁺ re-nucleation, ultimately leading to the uncontrollable Zn dendrite growth (*Angew. Chem. Int. Ed.* **61**, e202114789, doi:https://doi.org/10.1002/anie.202114789 (2022)). However, in our design, the energy barrier for Zn electrodeposition on various regions of Sb/Sb₂Zn₃-HI surface is lower and uniform, which is concluded from the uniform and larger current density distribution on the Sb/Sb₂Zn₃-HI surface. Therefore, Zn²⁺ ions can easily diffuse on the Sb/Sb₂Zn₃-HI@Cu surface. As a result, Zn nucleation on the Sb/Sb₂Zn₃-HI@Cu surface will occur over the entire area, and thus leading to uniform subsequent Zn deposition.

We have conducted in-situ observations of Zn deposition on Sb@Cu and Zn surfaces using optical microscopy to further verify their Zn electrodeposition behaviors. As shown in Figure R6, Zn deposition on the Zn foil in 2 M ZnBr₂ shows a nonuniform morphology in the initial Zn plating (5 s), and this inhomogeneity persists as the Zn deposition time increases (25-500 s). After deposition up to 1000 s, some obvious bumps were formed and gradually increased with time. The inhomogeneity of Zn²⁺ nucleation at the early stage of deposition on the Zn foil persisted throughout the deposition process and eventually formed uncontrollable dendrite growth. In contrast, the Zn²⁺ electrodeposition on the Sb/Sb₂Zn₃-HI@Cu surface shows a homogeneous morphology, which is attributed to the uniform electric field distribution and the homogeneous Zn nucleation at the initial

stage (5 s). In the following plating time (25-2000 s), the homogeneous electric field kept at the Sb/Sb₂Zn₃-HI interface and thus driving a uniform galvanized Zn deposition layer. The figures were added in the revised supporting information (Supplementary Fig. 10) and the discussion was updated in the revised manuscript. The updated contents were highlighted in the revised manuscript and also specified as follows:

(Page 10) *In-situ observation of Zn electrodeposition morphology on Zn foil and Sb@Cu substrate in 2 M ZnBr₂ further demonstrates the effect of Sb/Sb₂Zn₃-HI on stable Zn electrodeposition (Supplementary Fig. 10). Zn plating on the Zn foil shows a nonuniform morphology in the initial stage (5 s), and this inhomogeneity persists as the Zn deposition time increases (25-500 s). After deposition up to 1000 s, some obvious bumps were formed and gradually increased with time. In contrast, the Zn²⁺ electrodeposition on the Sb/Sb₂Zn₃-HI@Cu surface shows a homogeneous morphology, which is maintained throughout the entire galvanizing process (5-2000 s). It can be attributed to the homogeneous electric field kept at the Sb/Sb₂Zn₃-HI and thus driving a uniform galvanized Zn deposition layer.*

Figure R6. In-situ observation of morphologies of the Zn electrodeposition on Zn foil and Sb@Cu substrates in 2 M ZnBr₂ at a current density of 20 mA cm⁻².

The diffusion behaviors of Zn²⁺ on the Sb/Sb₂Zn₃-HI.

DFT calculations shows that the Sb surface has a lower adsorption energy than the Zn surface in the initial adsorption state. After the formation of a robust Sb/Sb₂Zn₃-HI with the increase of Zn atoms, the subsequently adsorbed Zn atoms always maintain the lower adsorption energy than the Zn surface. This suggests that the Sb/Sb₂Zn₃-HI is beneficial to homogeneous Zn plating. In addition, we conducted new experiment to calculate the diffusion behavior of Zn²⁺ on the Sb/Sb₂Zn₃-HI surface through DFT. Figure R7 shows the optimized structures with different adsorbed Zn numbers

on Sb surface. The geometric information clearly present that a stable Sb/Sb₂Zn₃ interface is gradually formed with the increase of the adsorbed Zn number. We further calculated the structure of Sb adsorbed with 21 Zn atoms and its structural configuration indicates that the formed Sb/Sb₂Zn₃ interface layer is more stable as the number of adsorbed atoms increases. Furthermore, based on the above optimized structures, we carried out the climbing image nudged elastic band (CI-NEB) calculations to investigate the Zn²⁺ diffusion process. We firstly tried to perform energy barrier calculations for Zn²⁺ diffusion inside the Sb/Sb₂Zn₃-HI, and the results showed relatively high values of 0.24 ~ 0.45 eV (Figure R8). In contrast, the diffusion process of Zn²⁺ at the formed stable Sb/Sb₂Zn₃ interface shows a lower energy barrier of 0.12 eV, indicating that Zn²⁺ diffuses much easier at the interface than inside the Sb/Sb₂Zn₃-HI (Figure R9). Therefore, we concluded that Zn²⁺ tend to migrate on the Sb/Sb₂Zn₃-HI interface, which is consistent with our experimental observation. The geometric information was added in the supporting information (Supplementary Fig. 4e) and the discussion was included in the revised manuscript. The updated contents were highlighted in the revised manuscript and also specified as follows:

(Page 8) *In addition, the geometric information clearly present that a stable Sb/Sb₂Zn₃ interface is gradually formed with the increase of the adsorbed Zn number (Supplementary Fig. 4e). We further calculated the structure of Sb adsorbed with 21 Zn atoms and its structural configuration indicates that the formed Sb/Sb₂Zn₃ interface layer is more stable as the number of adsorbed atoms increases. This indicates that Zn deposition on Sb/Sb₂Zn₃-HI is easy and verifies the positive effect of the Sb/Sb₂Zn₃ interfacial layer on the homogeneous Zn electrodeposition.*

Figure R7. Optimized structures with the different number (1~11, 21) of adsorbed Zn on the Sb (104).

Figure R8. The energy barrier and the corresponding structures for the diffusion process of Zn^{2+} inside the Sb/Sb₂Zn₃-HI.

Figure R9. Energy barrier and corresponding structure of the diffusion process of Zn^{2+} on $\text{Sb}/\text{Sb}_2\text{Zn}_3\text{-HI}$.

Comment 3: There are no equivalent circuits in the Nyquist plot of Figure S6. All the EIS data in the manuscript should be fitted and the corresponding equivalent circuit and fitting parameters should be provided.

Our response: Thanks so much for the helpful comment. We fitted the impedance data of the substrate in Figure S6, and the equivalent circuit is shown in Figure R1, where R_s is the ohmic resistance of solution and electrodes, R_{ct} is the charge transfer resistance, C_F is the double-layer capacitance, and Z_W is the Warburg impedance. We have added the equivalent circuit (Figure R1), fitted curves (Figure R2) and the corresponding fitting results (Table R2) in the revised supplementary information and updated the descriptions in the revised manuscript. The updated contents were highlighted in the revised manuscript and also specified as follows:

(Page 8) *As illustrated in Supplementary Fig. 6, 7 and Table 2. The $\text{Sb}@Cu$ substrate displays an ohmic resistance of 1.91Ω , which is lower than 2.56Ω of the Zn foil. Meanwhile, it also exhibits a lower charge transfer resistance of 2.62Ω than 19.78Ω of the Zn foil, indicating that the low adsorption energy on the Sb surface can effectively accelerate the charge transfer.*

Figure R1. An equivalent circuit is used to simulate the resistances of different substrates, where R_s is the ohmic resistance, R_{ct} is the charge transfer resistance, C_F is the double-layer capacitance, and Z_W is the Warburg impedance.

Figure R2. Electrochemical impedance spectroscopy of the Zn, Cu and Sb@Cu substrates and the corresponding fitting curves in 2 M $ZnBr_2$.

Table R2. The fitting results of EIS plots of all the samples in 2 M $ZnBr_2$.

Samples	R_s	R_{ct}	C_F
Sb@Cu	1.91	2.62	0.017
Cu	5.82	4.99	5.55e-5
Zn	2.56	19.78	3.06e-5

Comment 4: The high energy density of $\sim 274 \text{ Wh kg}^{-1}$ in this study is based on the mass of active materials. It is quite misleading. The total mass of the devices (including substrate, shell, and separator) should take into account for the evaluation of energy densities, which is more meaningful for practical applications.

Our response: We would like to thank the reviewer for the valuable comment. We agree with the reviewer's comment that the total mass of the battery device including every component should be taken into account for the evaluation of energy density. However, the report of energy density based on the mass of active materials of both cathode and anode provides an approach on the calculation of potential energy density of specific battery designs, which is also meaningful for the analysis of the battery chemistry.

As per the request of the reviewer, we have further optimized the scaled-up Zn-Br₂ battery under real conditions for practical energy density calculation. The 500 mAh scaled-up battery was built in a pouch cell, where the electrode sizes were about 4.5 cm * 3.5 cm with an areal capacity of 31.7 mAh cm⁻². Specifically, the optimized scaled-up battery consists of carbon felt cathode, Sb@Cu anode, electrolyte, and soft packaging. The total weight of the battery is about 12.4 g (Figure R10). The N/P ratio of the battery is about 1.25:1, and the electrolyte is used to wet the separator and electrodes. The charge and discharge curves of the pouch cell with the obtained discharge energy of 766.9 mWh are shown in Figure R11. Therefore, the battery shows an attractive energy density of about 62 Wh/kg based on the total mass of the whole pouch cell. We believe that the energy density of our battery can be largely increased upon further optimization in future study.

Figure R10. A digital photo of the weight of the pouch cell measured by an electric microbalance.

Figure R11. Charge and discharge curves of the pouch cell for energy density calculation.

Comment 5: Zn is highly reactive and forms SEI layers, and the competing surface reactions lead to complex "mossy" growth below the diffusion limited current (steady state) or before "Sand's time" (for galvanostatic over-limiting current). The following paper visually and electrochemically demonstrates the transition between mossy and dendritic growth of lithium at "Sand's capacity": Bai, Peng, et al. "Transition of lithium growth mechanisms in liquid electrolytes." *Energy & Environmental Science* 9.10 (2016): 3221-3229. The authors must check what is Sand's capacity for their battery systems and explain whether their experiments are carried out in the dendritic growth regime. Bai showed that many papers on lithium full cells that claim to "solve the dendrite problem" in fact are operating below Sand's capacity where dendrites would not occur anyway. This means the current was small or the time was short (too small capacity).

Our response: Thanks so much for the reviewer's insightful comment, which can help us to further understand the mechanism of Zn deposition by the heterogenous interface layer design. We have carefully studied the paper the reviewer mentioned above and designed a similar device for in-situ Zn electrodeposition measurements. The digital photo of the device is shown in Figure R12.

Figure R12. Digital photo of the device for the in-situ observation of Zn electrodeposition.

To investigate the "mossy" growth phenomena and search for "Sand" capacity of the Zn electrodeposition, we performed in-situ morphological observations of Zn electrodeposition on Zn foil and Sb@Cu substrates in 2 M ZnBr₂ at different current densities, including 3 mA cm⁻² and 20

mA cm⁻². The Sand time calculation formula mentioned in the literature (*Energy Environ. Sci.* **9**, 3221-3229, doi:10.1039/C6EE01674J (2016).) is shown in Equation R1:

$$t_{Sand} = \pi D_{app} \frac{(z_c c_0 F)^2}{4 (J t_a)^2} \quad \text{Equation R1}$$

where z_c is the charge number of the cation ($z_c = 2$ for Zn²⁺), c_0 is the bulk salt concentration of 2000 mol m⁻³, F is the Faraday's constant, J is the current density, and the D_{app} in 2 M ZnBr₂ is about 7×10^{-10} cm² s⁻¹. The transference number of Zn²⁺ was evaluated in a symmetric Zn cell with two Zn foils combined by EIS before and after the CA test in 2 M ZnBr₂ electrolyte (Figure R13), and calculated according to the following Equation R2 (*Adv. Energy Mater.* **12**, 2103705, doi:https://doi.org/10.1002/aenm.202103705 (2022).):

$$t_{Zn^{2+}} = \frac{I_s \times (\Delta V - I_0 R_0)}{I_0 \times (\Delta V - I_s R_s)} \quad \text{Equation R2}$$

where ΔV is the applied voltage polarization (10 mV in this work), I_s and R_s are the steady current and resistance, respectively, and I_0 and R_0 are the initial current and resistance, respectively. Thus, the $t_{Zn} = 0.48$ and $t_a = 1 - t_{Zn} = 0.52$ are the transference numbers of zinc cations and associated anions, respectively.

Based on Equation R1, when the applied current density is 3 mA cm⁻², the calculated Sand time is 3.36×10^5 s, which is 7567.5 s under the current density of 20 mA cm⁻².

Figure R13. Measurement of Zn^{2+} transference number, where the tests were conducted on a Zn|Zn symmetric cell with $80 \mu\text{L}$ 2 M ZnBr_2 electrolyte. (a) Nyquist plots of EIS at the initial and steady states. (b) The equivalent circuit model applied to fit the EIS (R_{es} : equivalent series resistance, R_i : interfacial resistance, R_{ct} : charge transfer resistance). (c) Current-time curves following DC polarization at 10 mV for the Zn|Zn cell.

As shown in Fig. R14a, Zn shows a dense galvanic layer at the initial deposition (5 s) and a "mossy" state when the deposition time reaches 5000 s , which gradually intensifies as the deposition time is further extended ($10000\text{-}36000 \text{ s}$). In contrast, the Zn electrodeposition on Sb@Cu surface does not appear "mossy", which maintains a homogeneous and dense state all along the plating duration (Figure R14b). In addition, the Zn electrodeposition morphology and potential versus time curves show that the "Sand" capacity is absent. We further increased the deposition current density (20 mA cm^{-2}) to expand the areal capacity of Zn deposition to observe if the "Sand" capacity would appear. As shown in Figure 15a, the "mossy" morphology of Zn electrodeposition rapidly appears on Zn rods ($5000\text{-}36000 \text{ s}$). However, even though the areal capacity has reached 200 mAh cm^{-2} ,

the "sand" capacity is consistently absent from the in-situ morphology and deposition curve observations. It is clear that the deposition time has far exceeded the theoretically calculated time (7567.5 s) for the appearance of the sand capacity. The Zn plating on the Sb@Cu rod maintains a uniform galvanization layer when the time is less than 20,000 s (Figure R15b). Even when Zn dendrites appear at the later stage of galvanization (more than 20,000 s), the potential curve remains without major fluctuations, indicating that the "sand" capacity never appears. It is concluded that our experiments are carried out in the dendritic growth regime, but the so-called "Sand" capacity is not observed. This is mainly due to the essential difference between Zn and Li electrodeposition in terms of different electrode materials, electrolyte system, cell structure, etc. Therefore, the Zn deposition cannot reach "Sand" capacity when the areal capacity is less than 200 mAh cm⁻². This ultrahigh capacity can fully support the practical application of Zn batteries and we therefore conclude that our strategy can be used to avoid Zn dendrite growth within a certain capacity (<200 mAh cm⁻²).

Figure R14. In-situ observations of the Zn electrodeposition on Zn and Sb@Cu rods in 2 M ZnBr₂ with a current density of 3 mA cm⁻². (a) Zn rod. (b) Sb@Cu rod.

Figure R15. In-situ observations of the Zn electrodeposition on Zn and Sb@Cu rods in 2 M ZnBr₂ with a current density of 20 mA cm⁻². (a) Zn rod. (b) Sb@Cu rod.

Comment 6: The authors are encouraged to evaluate the dendrite control effect of Sb/Sb₂Zn₃-HI in practical cell conditions, e.g. lean electrolyte or low N/P ratio, to validate the feasibility of this strategy in industrial level batteries.

Our response: We would like to thank the reviewer for the helpful suggestions. We agree with the reviewer that the evaluation of the growth of the Zn dendrites under practical cell conditions is very important to develop a practical battery system. Therefore, we conducted new experiment to evaluate the dendrite control effect of Sb/Sb₂Zn₃-HI in the optimized scaled-up battery. The 500

mAh scaled-up battery was built in a pouch cell, where the electrode size was about 4.5 cm * 3.5 cm with an areal capacity of 31.7 mAh cm⁻² and the N/P ratio of the battery is about 1.25:1 (Figure R10). We observed the morphologies of the Sb@Cu electrode at the charge and discharge states. As shown in Figure R16, the Sb@Cu substrate shows an overall flat and uniform morphology in the charge state, and no dendrites appear at the microscopic scale. When the cell is discharged, the Sb@Cu substrate remains in a homogeneous form. Some residual deposited Zn can be seen here because the N/P ratio is slightly higher than 1, but they are still in a homogeneous state without any dendritic appearance. All in all, our designed Sb/Sb₂Zn₃-HI in the scale-up battery effectively avoids the growth of Zn dendrites, showing its great opportunity for the industrial level battery.

Figure R16. Morphology of Sb@Cu substrates in the charged and discharged states in the optimized scale-up battery.

Reviewer #2 (Remarks to the Author):

General comment: This article reported an Sb/Sb₂Zn₃ heterostructured interface to regulate Zn nucleation and growth in Zn batteries. Experimental and theoretical investigations are implemented to unveil the Zn deposition mechanism. This finding is interesting, and this article deserves publication in Nature communication after answering the following questions:

Our response: We would like to thank the reviewer so much for his/her careful reading and positive comments, which will help improve the quality of our manuscript. Based on the reviewer's insightful suggestions, the manuscript has been thoroughly revised and all changes are highlighted in the revised manuscript.

Comment 1: In line 184, the authors claimed, "This suggests the uniform Zn plating can be well maintained even for thick Zn deposits." It is hard to get such a firm conclusion from the DFT results. In the Figure 2a, we could only get the adsorption energy versus the Zn number without the geometric information.

Our response: Thanks so much for the reviewer's good comment. We have changed the description here to make it more accurate. Moreover, we added the geometric information in the supporting information (Figure S4e) as shown in Figure R1. The geometric information clearly present that a stable Sb/Sb₂Zn₃ interface is gradually formed with the increase of the adsorbed Zn number. We further calculated the structure of Sb adsorbed with 21 Zn atoms and its structural configuration indicates that the formed Sb/Sb₂Zn₃ interface layer is more stable as the number of adsorbed atoms increases. This further verifies the positive effect of the Sb/Sb₂Zn₃ interfacial layer on the homogeneous Zn electrodeposition. The geometric information and above discussion have been updated in the revised manuscript and supporting information (Figure S4e) and specified as follows:

(Page 8) *In addition, the geometric information clearly present that a stable Sb/Sb₂Zn₃ interface is gradually formed with the increase of the adsorbed Zn number (Supplementary Fig. 4e). We further calculated the structure of Sb adsorbed with 21 Zn atoms and its structural configuration indicates that the formed Sb/Sb₂Zn₃ interface layer is more stable as the number of adsorbed atoms increases. This indicates that Zn deposition on Sb/Sb₂Zn₃-HI is easy and verifies the positive effect of the Sb/Sb₂Zn₃ interfacial layer on the homogeneous Zn electrodeposition.*

Figure R1. Optimized structures with the different number (1~11, 21) of adsorbed Zn on the Sb (104).

Comment 2: In the Figure 2c, it seems that there are Sb atoms around adsorbed Zn atoms, which explains the claim in line 187 "deposited Zn atoms on the Sb surface are more delocalized than that on the Zn surface." Why is the adsorption configuration different between Zn (001) and Sb (104)?

Our response: We would like to thank the reviewer for this comment. We displayed the optimized structures of the first Zn atom adsorption on Sb (104) surface in Figure 2c. As Zn and Sb undergo alloying at the initial stage of Zn adsorption on the Sb (104) surface, the adsorbed Zn atoms are surrounded by Sb atoms. Specifically, the initial adsorption of Zn atoms on the Zn foil surface is a homogeneous adsorption, so that the adsorbed Zn atoms appear on the surface of the Zn atomic layer. While the Zn atoms adsorbed on the Sb surface alloy with Sb to form a new Zn/Sb alloy phase, the Zn atoms appear to be surrounded by Sb atoms. Thus, Zn adsorption at different interfaces has different interatomic combination forms, which leads to different adsorption configurations.

Comment 3: For the phase field analysis, please add the scale in Figure 2d. Also, please tabulate the physical parameters.

Our response: Thanks so much for review's helpful suggestion. We have added scale bars in Figures 2d, 2e and S8 and provided a table of physical parameters for the simulation in Supplementary Table 3. The updated figures and table are shown in Figure R2, 3 and Table R1.

Figure R2. (d) Simulated current density distribution on Zn substrate. (e) Simulated current density distribution on Sb/Sb₂Zn₃-HI@Cu substrate.

Figure R3. Simulated Zn^{2+} flux distribution at different Zn electrodeposition durations. (a) Zn electrodeposition on Zn foil. (b) Zn electrodeposition on Sb/Sb₂Zn₃-HI@Cu substrate.

Table R1. The physical parameters for COMSOL simulation.

Parameters	Values	Descriptions
cbZn	2000 mol/m ³	Bulk concentration of Zn
rhoZn	7140 kg/m ³	Density of Zn
DZn	7e-10 m ² /s	Diffusion coefficient of Zn
MZn	0.06538 kg/mol	Molar mass of Zn
T	293.15 K	Temperature

Comment 4: In the Figure S7, why the Zn flux in Sb/Sb₂Zn₃ base is two orders less than in pure Zn base?

Our response: Thanks a lot for the helpful comment. Due to the inhomogeneous electric field distribution on the Zn substrate, the reactivity of Zn²⁺ varies greatly in different regions of the Zn foil surface, resulting in large variations in the Zn flux. Specifically, the Zn flux is about 1e-7 in the region where the Zn dendrites are formatted, while it remains at the level of 1e-9 in other regions. The Sb/Sb₂Zn₃-HI shows a uniform electric field distribution, thus the reactivity of Zn²⁺ in different regions of Sb/Sb₂Zn₃-HI is similar, resulting in a uniform Zn flux whose values are all in the order of 1e-9. As a result, the maximum value of the Zn flux on the Zn substrate surface is two orders of magnitude higher than that of the Sb@Cu substrate, which agrees well with the results in Figure S7.

Reviewer #3 (Remarks to the Author):

In this manuscript, the authors designed a robust antimony/antimony-zinc alloy heterostructured interface (Sb/Sb₂Zn₃-HI) to stabilize Zn nucleation and electrodeposition. Benefiting from the effects of stronger adsorption and homogeneous electric field distribution of the Sb/Sb₂Zn₃-HI in Zn plating, the dendrite-free Zn electrodeposition was achieved even at an ultra-high areal capacity of 200 mAh cm⁻². The realization of such a record high areal capacity of Zn plating is truly a breakthrough, offering the Zn-free Sb@Cu anodes a greater practical value. I have never seen from previous reports on such a high areal capacity of Zn anode. The authors then conducted various materials characterization, DFT calculation and simulation to reveal the mechanism of the anode-free Zn electrodeposition by the HI design. Furthermore, the anode-free Zn-Br₂ battery based on the Sb/Sb₂Zn₃-HI@Cu anode exhibited a high energy density and good cycling stability at an areal capacity of 10 mAh cm⁻². The authors further scaled the Zn-Br₂ battery up to Ah level, which displayed high energy density for the practical battery storage module by charging from photovoltaic cells. Overall, the anode material reported in this manuscript is very innovative and the achieved Ah-level batteries offer promises for the practical application of Zn batteries. This work provides an important advancement for the future design of Zn anode. Therefore, I strongly support this manuscript to be accepted after some minor revisions. The specific comments are as follows.

Our response: We would like to thank the reviewer for his/her positive comments and valuable suggestions. The manuscript has been thoroughly revised as per the reviewer's suggestions. All changes were highlighted in red color in the revised manuscript.

Comment 1: In the introduction, the authors mentioned that "Using well-engineered zincophilic substrates as anode current collectors can effectively inhibit the formation of Zn dendrites, while making them feasible to achieve anode-free Zn battery design". The zincophilic substrates are a broad concept, the authors need to list specific materials or categories to further inform the reader about the zincophilic materials.

Our response: Thank the reviewer so much for the helpful comment. We have provided a detailed list of common zincophilic materials in the revised manuscript. The updated contents with relevant papers citation were highlighted in the revised manuscript and specified as follows.

(Page 3) *Using well-engineered zincophilic materials such as Sn, Pb, Au, Ag, Cu and C on anode current collectors can effectively inhibit the formation of Zn dendrites, while making them feasible to achieve anode-free Zn battery design²⁷⁻³³.*

The corresponding references:

27. Weber R., et al. Long cycle life and dendrite-free lithium morphology in anode-free lithium pouch cells enabled by a dual-salt liquid electrolyte. *Nat. Energy* **4**, 683-689 (2019).
28. Cohn A. P., Muralidharan N., Carter R., Share K. & Pint C. L. Anode-free sodium battery through in situ plating of sodium metal. *Nano Lett.* **17**, 1296-1301 (2017).
29. Yin Y., et al. Dendrite-free zinc deposition induced by tin-modified multifunctional 3D host for stable zinc-based flow battery. *Adv. Mater.* **32**, 1906803 (2020).
30. Cui M., et al. Quasi-isolated Au particles as heterogeneous seeds to guide uniform Zn deposition for aqueous zinc-ion batteries. *ACS Appl. Energy Mater.* **2**, 6490-6496 (2019).
31. Zhang Y., Howe J. D., Ben-Yoseph S., Wu Y. & Liu N. Unveiling the origin of alloy-seeded and nondendritic growth of Zn for rechargeable aqueous Zn batteries. *Acs Energy Lett.* **6**, 404-412 (2021).
32. Xie S., et al. Stable zinc anodes enabled by zincophilic Cu nanowire networks. *Nano-Micro Lett.* **14**, 39 (2021).
33. Zhu Y., Cui Y. & Alshareef H. N. An anode-free Zn–MnO₂ battery. *Nano Lett.* **21**, 1446-1453 (2021).

Comment 2: In the “Design of 2D Heterostructured Interface” section, the authors mentioned that “Cu was selected as the substrate for the Zn-free anode to build up a durable Sb@Cu electrode (Supplementary Fig. 1)”. Here the authors should add the explanation of the mechanism on why Sb and Cu can form a durable interfacial layer.

Our response: We would like to thank the reviewer for the helpful suggestions. We have extended the explanation of the mechanism on the design of heterostructured interface in the revised manuscript. The updated contents were highlighted in the revised manuscript and specified as follows:

(Page 5) *As illustrated in Supplementary Fig. 1, a typical Cu₂Sb phase was detected in the XRD results, which indicates that Sb has a certain solubility in Cu, promoting a strong bond between the two metals and thus establishing a robust Sb-Cu interface⁴⁵.*

Comment 3: The rate capacities shown in Figure 4c do not match with the description. The authors need to further check and update this part.

Our response: Thanks for the reviewer's helpful question. We checked Figure 4c and corrected it in the revised manuscript. The specific update of the Figure 4c is shown as below.

Figure 4c. Rate capacities of the Zn|Sb@Cu half-cell at an areal capacity of 10 mAh cm⁻².

Comment 4: The authors performed a Zn-Br₂ battery cycle performance test using only Sb@Cu anodes. Additional Zn-Br₂ battery cycle performance tests using Zn foil as the anode are needed for better comparison.

Our response: Thanks so much for the reviewer's insightful comment. The cycling performance of Zn-Br₂ batteries is indeed very important to demonstrate their potential for large-scale energy storage applications. Meanwhile, the cycling performance test of Zn-Br₂ battery does require the addition of Zn foil anodes as a comparison to further confirm the advantages of Sb@Cu substrates. We have added the cycling performance of Zn-Br₂ battery using a Zn foil anode. Figure 5d and the corresponding description were highlighted in the revised manuscript and shown below.

(Page 15) *In contrast, the battery with Zn foil anode can only be cycled steadily for about 50 cycles, followed by a rapid drop in efficiency and eventually complete failure in less than 100 cycles. This can be attributed to the uncontrollable Zn dendrite growth resulted short-circuiting of the battery.*

Figure 5d. Long-term cycling performance at 10 mAh cm⁻² and 10 mA cm⁻².

Comment 5: The electrolyte will affect of the electrochemical performance of energy storage systems. You can do some corresponding research to improve the stability of your battery, at least cite some relevant papers to give a potential promotion methods, just example: Nat. Sustain., 2022, 5, 225-234; DOI: 10.1002/anie.202208291; National Science Review, nwac134.

Our response: Thanks so much for the reviewer's good suggestion. We agree with the reviewer that the electrolyte is an important part of the battery and reasonable adjustment of the electrolyte is a wise strategy to improve the stability the Zn-Br₂ battery. However, this work is mainly focused on the study of anode-free Sb@Cu substrates, and thus the systematic study on electrolyte optimization will be carried out in our future studies. We have cited the mentioned three papers and highlighted them in the revised manuscript, which corresponds to the references 24, 25 and 26. The specific updates are listed as below:

24. Li J., et al. Weak cation–solvent interactions in ether-based electrolytes stabilizing potassium-ion batteries. Angew. Chem. Int. Ed. 61, e202208291 (2022).

25. Ge J., Fan L., Rao A. M., Zhou J. & Lu B. Surface-substituted prussian blue analogue cathode for sustainable potassium-ion batteries. Nat. Sustain. 5, 225-234 (2022).

26. Hu Y., et al. Cyclic-anion salt for high-voltage stable potassium-metal batteries. Natl. Sci. Rev. 9, nwac134 (2022).

Comment 6: The authors carried out a scale-up study of Zn-Br₂ battery, showing good Coulombic efficiency. But its cycling stability is still much worse than that of small Zn-Br₂ battery (800 cycles), what are the key issues for the enlarged Zn-Br₂ battery?

Our response: Thanks for the reviewer's insightful comment. The excellent electrochemical performance of the scaled-up Zn-Br₂ battery is important to large-scale energy storage applications. The cycling performance of the scaled-up Zn-Br₂ battery is shown in Figure 7c, which displays a relatively good stability for over 400 cycles. In a scaled-up battery, the electrochemical performance is not only influenced by the side reactions at the electrodes (anode: Zn dendrites, hydrogen evolution reaction, corrosion; cathode: dissolution, passivation), but also by the cell structure, the electrode/current collector connection, and the battery assembly form. Our future work will be focused on developing integrated strategies to solve the specific problems and to further optimize the scaled-up Zn-Br₂ battery for large scale energy storage applications.

Comment 7: The authors reported an attractive energy density of the Zn-Br₂ battery (about 274 Wh/kg), which was based on the active materials of anode and cathode. Did the authors evaluate the energy density of the entire scaled-up battery? This should include the active materials, current collectors, electrolyte, etc. It will help to demonstrate the practical application of the battery.

Our response: We would like to thank the reviewer for the valuable comment. We agree with the reviewer's comment that the total mass of the battery device including every component should be taken into account for the evaluation of energy density.

As per the request of the reviewer, we have further optimized the scaled-up Zn-Br₂ battery under real conditions for practical energy density calculation. The 500 mAh scaled-up battery was built in a pouch cell, where the electrode sizes were about 4.5 cm * 3.5 cm with an areal capacity of 31.7 mAh cm⁻². Specifically, the optimized scaled-up battery consists of carbon felt cathode, Sb@Cu anode, electrolyte, and soft packaging. The total weight of the battery is about 12.4 g (Figure R1). The N/P ratio of the battery is about 1.25:1, and the electrolyte is used to wet the separator and electrodes. The charge and discharge curves of the pouch cell with the obtained discharge energy of 766.9 mWh are shown in Figure R2. Therefore, the battery shows an attractive energy density of about 62 Wh kg⁻¹ based on the total mass of the whole pouch cell. We believe that the energy density of our battery can be largely increased upon further optimization in future study.

Figure R1. A digital photo of the weight of the pouch cell measured by an electric microbalance.

Figure R2. Charge and discharge curves of the pouch cell for energy density calculation.

REVIEWERS' COMMENTS

Reviewer #1 (Remarks to the Author):

The authors did a good work in the revision of the manuscript, and my concerns have been fully addressed. I recommend the publication of this paper in Nature Communications.

Reviewer #2 (Remarks to the Author):

The author's responses are clear and accurate. Their responses answered our questions reasonably. We do not have any more questions or suggestions about this manuscript. Therefore, we propose to publish this manuscript in Nature Communications.

Reviewer #3 (Remarks to the Author):

The authors have addressed all my concerns. I recommend its publication in Nature Communications.

Response to Reviewers' Comments

We thank the reviewers for their valuable comments. We are delighted that all three reviewers have recommend our manuscripts for publication in Nature Communications. For ease of reference, the reviewers' comments and suggestions are reproduced in **blue**, our response is in **black**.

Reviewer #1 (Remarks to the Author):

The authors did a good work in the revision of the manuscript, and my concerns have been fully addressed. I recommend the publication of this paper in Nature Communications.

Our response: We would like to thank the reviewer for your thorough reading and positive feedback.

Reviewer #2 (Remarks to the Author):

The author's responses are clear and accurate. Their responses answered our questions reasonably. We do not have any more questions or suggestions about this manuscript. Therefore, we propose to publish this manuscript in Nature Communications.

Our response: Thanks so much for recommending our articles for publication in Nature Communications.

Reviewer #3 (Remarks to the Author):

The authors have addressed all my concerns. I recommend its publication in Nature Communications.

Our response: Thank you very much for your recognition and recommendation of our work.